# Asymmetric conformations and lipid interactions shape the ATP-coupled cycle of a heterodimeric ABC transporter

Qingyu Tang [1], Matt Sinclair [2,3], Hale S. Hasdemir [2,3], Richard A. Stein [1], Erkan Karakas [1], Emad Tajkhorshid [2] & Hassane S. Mchaourab [1] ✉

Here we used cryo-electron microscopy (cryo-EM), double electron-electron resonance spectroscopy (DEER), and molecular dynamics (MD) simulations, to capture and characterize ATP- and substrate-bound inward-facing (IF) and occluded (OC) conformational states of the heterodimeric ATP binding cassette (ABC) multidrug exporter BmrCD in lipid nanodiscs. Supported by DEER analysis, the structures reveal that ATP-powered isomerization entails changes in the relative symmetry of the BmrC and BmrD subunits that propagates from the transmembrane domain to the nucleotide binding domain. The structures uncover asymmetric substrate and $Mg^{2+}$ binding which we hypothesize are required for triggering ATP hydrolysis preferentially in one of the nucleotide-binding sites. MD simulations demonstrate that multiple lipid molecules differentially bind the IF versus the OC conformation thus establishing that lipid interactions modulate BmrCD energy landscape. Our findings are framed in a model that highlights the role of asymmetric conformations in the ATP-coupled transport with general implications to the mechanism of ABC transporters.

ATP-binding cassette (ABC) transporters harness the energy of ATP hydrolysis to traffic diverse substrates across cell membranes[1–3]. The remarkable breadth of their substrates confers to ABC transporters critical roles in physiological processes and in multidrug resistance[3–6]. Prokaryotic ABC transporters mediate nutrient uptake, antimicrobial peptide export, cell viability, virulence, and pathogenicity[7]. In humans, a number of ABC transporters of the exporter type are involved in lipid homeostasis, multidrug efflux, and the transport of immunogenic peptides. In addition, these transporters have been implicated in cardiovascular, metabolic, and neurodegenerative diseases[8–11].

Numerous studies have defined the basic molecular architectures of ABC exporters, delineated structural elements involved in their ATP-powered alternating access and revealed modes of substrate interactions[3,4,12–21]. Invariably, ABC transporters have a conserved architecture consisting of two nucleotide-binding domains (NBDs) which hydrolyze ATP to energize alternating access of two transmembrane domains (TMDs), the site of substrate recognition and translocation[3]. The NBDs are considered the hallmark of the family sharing extensive sequence similarity and architecture among ABC transporters[1]. In contrast, at least 7 types of ABC transporters identified based on TMD folds and topologies. Among them, type IV ABC exporters, whose members include P-gp, TmrAB, TM287/288, MsbA, MRP1, McjD, and ABCB4, share a domain-swapped TMD arrangement[22].

While there is an emerging consensus on the structural elements of alternative access of type IV ABC exporters, the mechanism of ATP energy coupling has been less characterized. One of the prominent models, initially deduced from biochemical analysis of ABCB1

[1]Department of Molecular Physiology and Biophysics, Vanderbilt University, Nashville, TN 37232, USA. [2]Theoretical and Computational Biophysics Group, NIH Resource for Macromolecular Modeling and Visualization, Beckman Institute for Advanced Science and Technology, Department of Biochemistry, and Center for Biophysics and Quantitative Biology, University of Illinois at Urbana-Champaign, Urbana, IL 61801, USA. [3]These authors contributed equally: Matt Sinclair, Hale S. Hasdemir. ✉e-mail: hassane.mchaourab@vanderbilt.edu

(P-glycoprotein, Pgp)[23–25], posits the alternating hydrolysis of ATP in the two nucleotide binding sites, despite Pgp possessing two active NBDs organized in the context of a single polypeptide chain, and DEER analysis of Pgp[26,27] yielded direct evidence supporting the population of locally asymmetric intermediates in the substrate- and ATP-coupled cycle. In contrast, cryo-EM structures of Pgp, in the absence of substrates, were found to be symmetric particularly at the nucleotide-binding sites (NBSs)[21]. This Pgp construct contained substitution of the catalytic glutamate thereby impairing ATP hydrolysis. Subsequent DEER studies demonstrated that asymmetry was abrogated by the substitution of the glutamates, by the absence of substrates as well as by binding to high affinity inhibitors[26] rationalizing the cryo-EM structures and suggesting a critical role for asymmetric conformations in the functional mechanism.

A remarkable variation on the theme of ATP energy conversion in ABC transporters is the evolution of heterodimeric ABC exporters[25,28] which are distinguished by having an intrinsically asymmetric ATP catalytic cycle resulting from the selective substitution of one of the catalytic glutamates of the Walker B motif in the NBD[13,29,30] (Supplementary Fig. 1). Thus, they provide unique models to investigate structural asymmetry and elucidate its consequence on the conformational cycle with relevance to ABC exporters in general. A number of heterodimeric ABC exporters have been investigated[13,30,31], yielding insight into their structures and the mechanism of alternating access. TmrAB, an ABC heterodimeric transporter from *Thermus thermophilus*, was captured in five conformations that were proposed to represent the ATP-powered cycle[13]. However, as noted previously by us[12] and others[25,32], the energetics of the TmrAB conformational cycle may not be reflective of the typical type IV exporter. In particular, the dynamics of the NBDs may be constrained by strong helix/helix interactions at the C-terminus. Moreover, the role of the lipid-transporter interactions was not explicitly explored. Finally, the quality of the experimental density for the peptide substrate in these structures was not amenable to detailed structure determination.

Our model heterodimeric ABC exporter, BmrCD, is presumed to be associated with drug efflux in the Gram-positive bacterium *Bacillus subtilis*[33]. BmrCD is a heterodimer consisting of two protomers: BmrC and BmrD. Incorporated in giant unilamellar vesicles, BmrCD can transport ethidium bromide in vitro, a process powered by ATP and inhibited by orthovanadate[34]. Previous studies from our laboratory revealed that ATP-coupled conformational changes in BmrCD entail population of asymmetric configurations of the catalytically inequivalent NBSs[29,35]. A cryo-EM structure of an ATP-bound inward-facing (IF) conformation of BmrCD in detergent micelles uncovered an asymmetric arrangement in the TMD that was ostensibly stabilized by the inequivalent binding of two molecules of the substrate Hoechst-33342 (hereafter referred to as Hoechst or HT)[12].

The stability of the ATP- and Hoechst-bound IF conformation of BmrCD was considered unusual, as most type IV ABC exporters were captured in an outward-facing (OF) conformation under these conditions[13,21,36–39]. Along with DEER analysis demonstrating the requirement of ATP hydrolysis to stabilize the OF conformation[29], an ATP- and HT-bound IF conformation suggests a distinct energy landscape for BmrCD, particularly in comparison to TmrAB. Furthermore, BmrCD's conformational equilibrium shows strong dependence on interactions with the lipid environment. Although ATP hydrolysis is stimulated by incorporation of BmrCD into lipid bilayers, the NBD to TMD coupling, which drives transition to the OF conformation, becomes strictly dependent on substrate binding[29]. Even in the presence of the substrate Hoechst, only a fraction of BmrCD reaches the OF conformation. Finally, a distinct feature of BmrCD sequence is the presence of an extracellular domain (ECD) that caps the TMD (Supplementary Fig. 1). Thus, BmrCD provides a unique model to dissect the energetics of lipid modulation of ATP-coupled transport in ABC transporters[29].

To address the outstanding questions of the role of asymmetry and lipid interactions in the conformational cycle of ABC exporters, we determine structures of BmrCD in multiple distinct conformations in lipid nanodiscs and correlated the population of these conformations with DEER analysis under different nucleotide and substrate conditions. This integrated analysis illuminates aspects of the BmrCD conformational cycle. First, the transition entails population of structurally asymmetric intermediates. Asymmetry propagates from the TMD (in the IF conformation) to the NBD (in the occluded (OC) conformation) leading to differential $Mg^{2+}$ binding at the two NBSs (Supplementary Fig. 1) and ATP hydrolysis at the consensus NBS. Second, we determined IF structures without the substrate Hoechst, with one, or two antiparallel Hoechst molecules. Importantly, we observed that Hoechst binding is associated with differential $Mg^{2+}$ binding at the NBSs. Finally, lipids shape the energy of the cycle through differential interactions with the IF and OF conformations. Lipid molecules, visualized from cryo-EM maps, were subjected to MD simulations to identify those that are stably bound. We find that the IF conformation of BmrCD, by virtue of exposure of critical residues to the bilayer, interacts more extensively with lipids, thus explaining the stabilization of IF relative to OF. These findings are integrated to propose a transport model for heterodimeric ABC transporters with implications to type IV exporters.

## Results
### IF and OC conformations of BmrCD in lipid bilayers
We determined cryo-EM structures of wild type (WT) and a catalytically impaired mutant of BmrCD in lipid nanodiscs under conditions expected to stabilize distinct transport intermediates. The mutant, constructed on a cysteine-less (C-less) background, contains glutamine substitutions at D500 of BmrC and E592 of BmrD that impair ATP hydrolysis (referred to hereafter as BmrCD*-QQ) (Supplementary Fig. 1). The purified proteins were reconstituted in lipid nanodiscs of two different compositions: phosphatidylcholine (PC)/phosphatidic acid (PA), and PC and *E. coli* polar lipids, previously shown to stimulate ATP hydrolysis in WT and C-less backgrounds[29]. Although we determined multiple structures of WT and QQ BmrCD in both types of nanodiscs (Supplementary Fig. 2 and Table 1), we focus here on the best-quality, unique multiple conformations which were obtained in the BmrCD*-QQ background.

**Table 1 | Conditions and Ligands of BmrCD cryo-EM structures**

| | Dataset | MSP | Lipids (No.) | Hoechst | ATP | Mg²⁺ | Construct |
|---|---|---|---|---|---|---|---|
| BmrCD_IF-2HT/ATP | 1 | MSP1D1 | PC/PA (20) | 2 | 2 | 1 | BmrCD*-QQ |
| BmrCD_IF-1HT/ATP | 2 | MSP1D1 | PC/*E.coli* polar lipid (21) | 1 | 2 | 1 | |
| BmrCD_IF-ATP | 3 | MSP1D1E3 | PC/PA (25) | 0 | 2 | 0 | |
| BmrCD_IF-ATP2 | 4 | MSP1D1 | PC/*E.coli* polar lipid (13) | 0 | 2 | 0 | |
| BmrCD_OC-ATP | 4 | MSP1D1 | PC/*E.coli* polar lipid (16) | 0 | 2 | 2 | |
| BmrCD_OC-ADPVi | 5 | MSP1D1 | PC/PA (21) | 0 | 2 | 2 | BmrCD-WT |

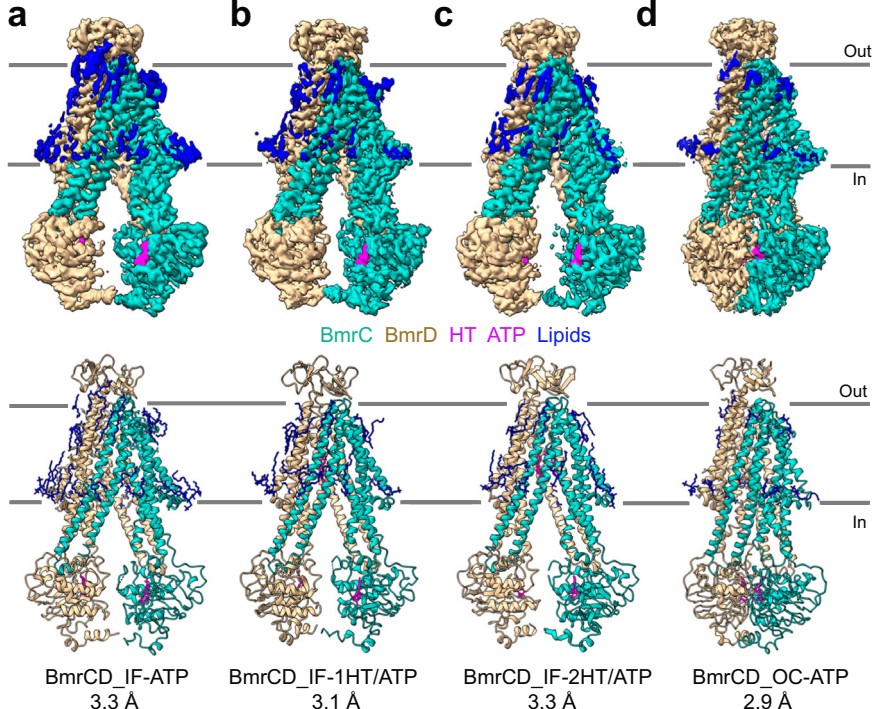

**Fig. 1 | Representative BmrCD structures in lipid bilayers. a** BmrCD inward-facing (IF) conformation bound to ATP. **b** BmrCD IF bound to one substrate Hoechst and two ATP molecules. **c** BmrCD IF conformation bound to two Hoechsts (HT) and two ATP molecules. **d** BmrCD occluded (OC) conformation bound to ATP. The top and bottom rows show experimental maps and deposited models, respectively. BmrC is shown in light sea green, BmrD is in tan, Hoechst (HT) and ATP are shown in magenta, and lipids are blue. In the models, BmrCD is shown in cartoon representation, and all the ligands are represented by sticks. $Mg^{2+}$ ions are omitted.

We report the structure of three IF conformations bound to different combinations of ATP and Hoechst. Under conditions of excess substrate Hoechst and ATP/$Mg^{2+}$, an IF conformation of BmrCD*-QQ in PC/PA lipids was determined to 3.3 Å resolution (referred to hereafter as BmrCD_IF-2HT/ATP) (Fig. 1c, Supplementary Figs. 2–6 and Supplementary Table 1). Along with 2 Hoechst and 2 ATP molecules, 20 lipid molecules can also be partially or fully modeled into the density (Supplementary Fig. 6). Remarkably only a single $Mg^{2+}$ ion in the consensus NBS was visible in the cryo-EM maps (see below for a detailed discussion). The BmrCD_IF-2HT/ATP structure is quite similar to the BmrCD structure in detergent micelles (PDB ID: 7M33, referred hereafter as BmrCD_det)[12] with an RMSD of 0.7 Å across 959 Cα atoms (Supplementary Table 2).

Under similar conditions of Hoechst and ATP/$Mg^{2+}$ but reconstituted in PC/*E. coli* polar lipids nanodiscs (cryo-EM dataset 2, labeled in Supplementary Fig. 2, Fig. 1b), another IF conformation of BmrCD*-QQ was identified with density for only one Hoechst molecule in the chamber (Fig. 1b, Supplementary Figs. 2–6 and Supplementary Table 1). This structure, referred to as (BmrCD_IF-1HT/ATP) represents an intermediate state with two ATP molecules, one $Mg^{2+}$ and one substrate. It appears that the energetics of BmrCD*-QQ in the PC/*E. coli* polar lipids mixture is favorable for an IF conformation bound to a single Hoechst molecule. Twenty-one lipid molecules were visualized in this structure.

The third IF conformation of BmrCD*-QQ was obtained in the presence of ATP but absence of Hoechst in PC/PA bilayers (referred hereafter as BmrCD_IF-ATP, Fig. 1a, Supplementary Figs. 2–6, and Supplementary Table 1). BmrCD_IF-ATP is characterized by a large chamber open to the cytoplasm, had two well resolved ATP molecules, although no identifiable $Mg^{2+}$ ion, as well as 25 densities that arise from lipid molecules (Supplementary Fig. 6). This IF structure is also very similar to BmrCD_IF-2HT/ATP and BmrCD_det, with

RMSD values of 0.9 Å across 1,153 Cα atoms and 1.4 Å across 1078 Cα atoms, respectively. However, in the absence of bound Hoechst, this IF conformation features a larger substrate binding cavity compared to the detergent and Hoechst-bound IF conformations. The solvent-accessible volume of BmrCD_IF-ATP substrate-binding chamber calculated by CASTp is 9,354 $Å^3$ whereas it is 7,882 $Å^3$ in BmrCD_IF-2HT/ATP and 7,538 $Å^3$ in BmrCD_det (Supplementary Fig. 7)[40]. One possible reason for the narrowing of the chamber in the later structures is interaction with the substrate Hoechst which can pull in the helices lining the chamber. However, it should be noted that BmrCD_IF-ATP was determined in larger nanodiscs using the MSP1D1E3 scaffold protein[41]. To exclude the possibility that the size of the nanodiscs influence the opening of the chamber, we examined ATP-bound BmrCD in nanodiscs formed using a smaller membrane scaffold protein MSP1D1, with a diameter of 9.5 nm compared to MSP1D1E3 of 12 nm (cryo-EM dataset 4, labeled in Supplementary Fig. 2). We identified a very similar IF structure (BmrCD_IF-ATP2) suggesting that BmrCD conformation is not constrained by the size of nanodiscs (Supplementary Fig. 2), a finding consistent with previous results by other groups[14].

In addition to the IF conformations, we obtained the structure of an occluded (OC) conformation of BmrCD*-QQ in PC/*E. coli* polar lipids (referred hereafter as BmrCD_OC-ATP, Fig. 1d, Supplementary Figs. 2–6, 8 and Supplementary Table 1, cryo-EM dataset 4). In this inward-closed structure, the NBDs are in contact with each other to form the NBSs and the chamber was occluded to the cytoplasm. In the same dataset, we also detected IF conformations in 2D classification and 3D models (Supplementary Figs. 2–6, 9). The cryo-EM maps displayed anisotropy in both OC and IF conformations (Supplementary Fig. 5d, e). Heterogeneous refinement implemented in CryoSPARC[42] was helpful in resolving the anisotropy problem of the maps for the IF and OC structures.

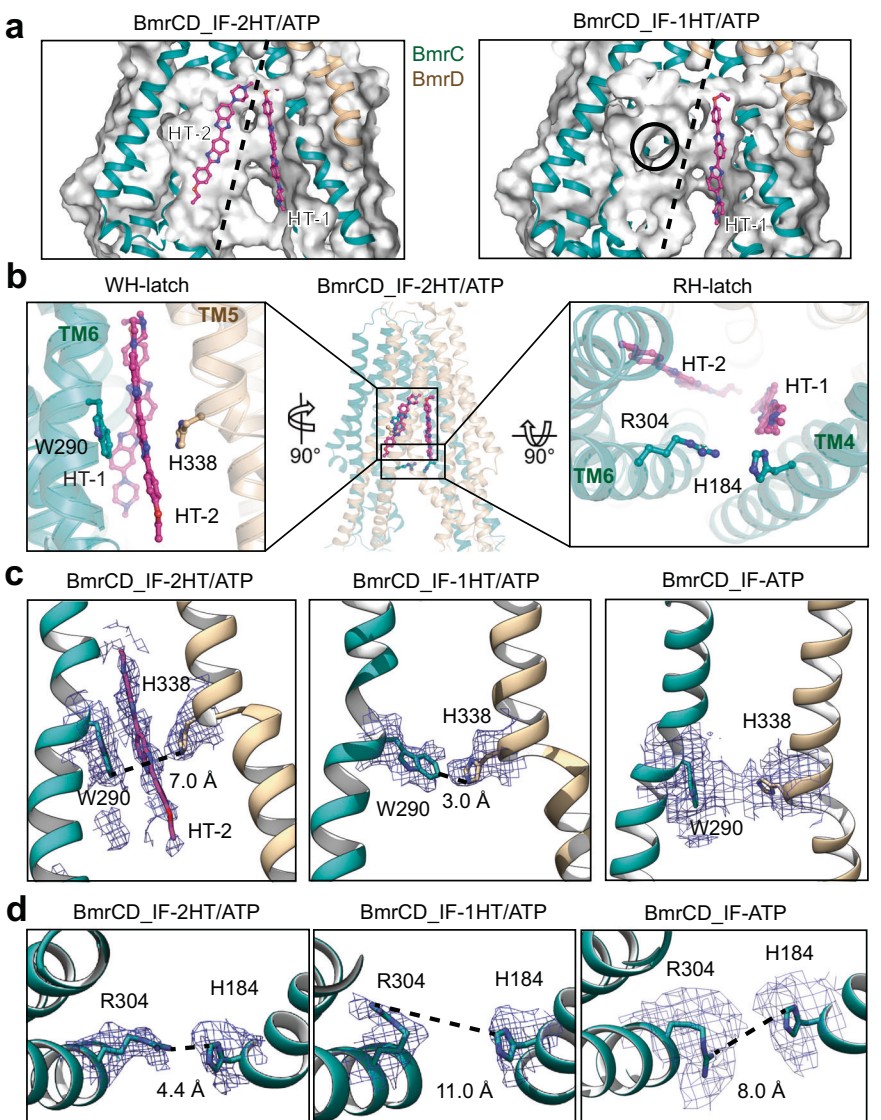

**Fig. 2 | Comparison of IF conformations with one or two bound Hoechst revealed that binding involve two latches in the substrate pocket. a** Side views of the surface of the substrate Hoechst-binding pocket of BmrCD_IF-2HT/ATP (left panel) and BmrCD_IF-1HT/ATP (right panel). The view is the same as the square box in Fig. 2b. BmrCD is shown in cartoon and Hoechst (HT, magenta color) is shown in stick. The dash black lines indicate two halves of the chamber accommodating two bound Hoechst molecules. The black circle in right panel highlights the second Hoechst-binding location. **b** The two latches locations are shown in BmrCD_IF-2HT/ ATP in the middle panel. The WH-latch is highlighted on in the left panel, and the RH-latch is highlighted on in the right panel. **c** WH-latch analysis. The densities for WH-latches in BmrCD_IF-2HT/ATP and BmrCD_IF-1HT/ATP are shown in the left and middle panels with the side-chain distances highlighted. The densities for WH-latches in BmrCD_IF-ATP is shown in right panel. **d** Analysis of the RH-latch in the absence and presence of one or two Hoechst molecules. The densities for RH-latches in BmrCD_IF-2HT/ATP (left panel), BmrCD_IF-1HT/ATP (middle panel) and BmrCD_IF-ATP (right panel) are shown with the side-chain distance highlighted.

Finally, while attempting to obtain an OF conformation in a WT background, we determined another OC structure in PC/PA bilayers (cryo-EM dataset 5). BmrCD was incubated with ATP and vanadate (Vi) to trap the high energy state (HES) populated following ATP hydrolysis (Supplementary Figs. 2–6). The OC in the WT and QQ backgrounds are almost identical (RMSD value of 0.6 Å over 1039 Cα atoms, Supplementary Table 2). Although we could not conclusively establish from the density, we presume that it likely had ADP and Vi at the consensus NBS, given a 10-min incubation at 37 °C. The cryo-EM maps of an IF conformation under these conditions was not amenable to high resolution structure primarily due to highly flexible NBDs. Overall, we didn't observe substantial differences for the same conformations between the two lipid compositions or between the WT and QQ OC conformations. Moreover, the stably bound lipid molecules, as identified by MD simulations (see below), appeared in similar locations in

all the structures regardless of lipid composition (Supplementary Fig. 10).

## A sequential Hoechst binding model involving two ionic latches

The two IF structures with one (BmrCD_IF-1HT/ATP) or two (BmrCD_IF-2HT/ATP) Hoechst molecules suggest a two-step binding process that entails changes in the interactions of two ionic latches in the substrate binding chamber. Superposition of the two structures highlight high similarity with a RMSD of 0.4 Å across 1,131 Cα atoms. Remarkably, in both IF structures one Hoechst (HT-1) molecule is located at the same position on the right half of the chamber (Fig. 2). As noted previously[12], the binding of the antiparallel Hoechst molecules is stabilized by two patches of negatively charged residues in orientations that match the bound Hoechst. A close-up analysis shows that W290 of BmrC and H338 of BmrD form an ionic latch (WH-latch) in the middle of the left

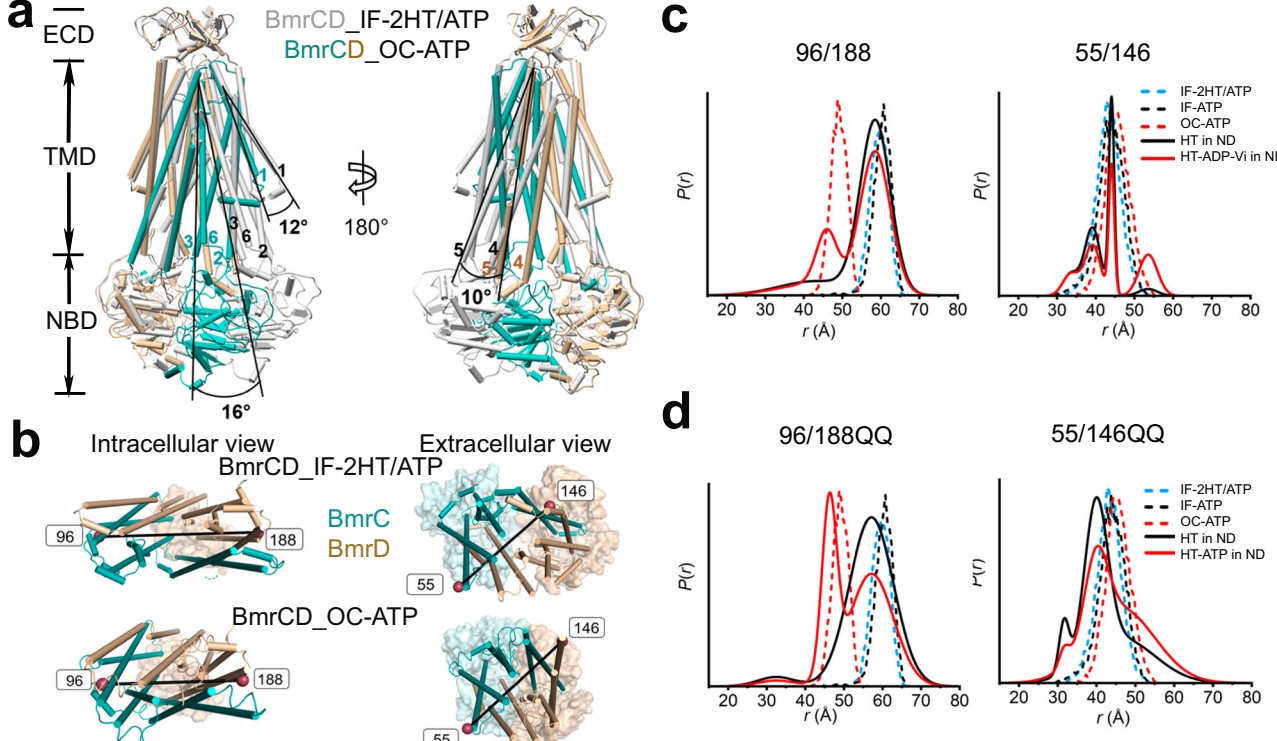

**Fig. 3 | Conformational changes underlying the IF to OC transition.**
**a** Superposition of BmrCD_IF-2HT/ATP and BmrCD_OC-ATP based on the alignment of BmrD. BmrC and BmrD in the OC conformation are shown in light sea green and tan, respectively while BmrCD_IF-2HT/ATP is shown in gray. TMs 1,2/3 and 6 of BmrC and 4/5 of BmrD undergo the largest shifts between the two conformations. Ligands are omitted in both structures for clarity. **b** Close up views of BmrCD highlighting spin label pairs 55$^{BmrC}$/146$^{BmrD}$ and 96$^{BmrC}$/188$^{BmrD}$. BmrC and BmrD are shown in light sea green and tan in BmrCD_OC-ATP, respectively. TMD helices are shown as cylinders and ECD and NBDs are shown as transparent surfaces. The spin label locations are highlighted by the spheres. **c, d** DEER distance measurements for spin-label pairs on the intracellular (96$^{BmrC}$/188$^{BmrD}$) and extracellular (55$^{BmrC}$/146$^{BmrD}$) sides in WT and QQ backgrounds respectively. Experimental distance distributions in nanodiscs representing the probability of a distance $P(r)$ versus the distance ($r$) between spin labels, are shown in solid lines: HT (refers to Hoechst hereafter, black) and HT-ADP-Vi (refers to Hoechst-ADP-vanadate hereafter, red). ND refers to nanodiscs hereafter. The simulated distance distributions derived from cryo-EM structures by MDDS are shown in dashed lines: IF-2HT/ATP (blue), IF-ATP (black), and OC-ATP (red).

---

part of the chamber in BmrCD_IF-1HT/ATP which blocks the binding of the second Hoechst (HT-2) in this structure (Fig. 2b, c, Supplementary Fig. 11). Distinct densities for the side chains of these residues unambiguously suggest their projection into the location of the second Hoechst. The latch is open in the fully loaded structure. The importance of W290 was highlighted in our previous work showing that ablation of its side chain eliminates cooperative simulation of ATP turnover by Hoechst[12].

Furthermore, examination of the chamber entrance at or around the intracellular side of the TMD exposes another ionic latch consisting of R304 and H184 of BmrC (RH-latch, Fig. 2d). This latch is open in the one-Hoechst structure with the sidechain distance at 11 Å (Fig. 2d, middle panel), whereas it is closed in the two-Hoechst structure with the sidechain distance at 4.4 Å (Fig. 2d, left panel). We propose that the energetics of these latches contribute to the overall binding of Hoechst and may well modulate the number of bound molecules. Interestingly, in the substrate-free IF structure BmrCD_IF-ATP, the RH-latch is open with the sidechain distance of 8 Å primed for the first Hoechst binding (Fig. 2d, right panel). The WH-latch opens for the second Hoechst which may then trigger closing of the RH-latch at the entrance.

## Structural rearrangements underlying the IF to OC transition of BmrCD
Superposition of BmrCD_IF-2HT/ATP and BmrCD_OC-ATP, yielded an RMSD value of 6.5 Å over 1,199 Cα atoms, suggesting drastic structural rearrangements. Comparison at the domain level highlights that the largest movements are mainly attributed to the TMDs with RMSD

values of 2.3 Å for BmrD-TMD (259 Cα atoms) and 4.8 Å for BmrC-TMD (314 Cα atoms). By contrast, the RMSD values at the NBDs are smaller: 1.3 Å for BmrD-NBD (190 Cα atoms) and 1.1 Å for BmrC-NBD (215 Cα atoms).

TM by TM inspection revealed that upon transition to the OC conformation, TM1 of BmrC tilts by 12° and the TM2/TM3 pair and TM6 of BmrC tilt by 16°. TMD rearrangement is coupled with the movement of the NBDs toward each other by ~23 Å to form two engaged NBSs. The transition between IF and OC closes the intracellular TMDs thereby occluding substrate entrance. Concomitantly, the TM4/TM5 pair of BmrD also tilts about 10° to contribute to closing of the entrance (Fig. 3a, Supplementary Fig. 7). A similar conformational change from IF to OC and OF structures also can be observed in the heterodimeric ABC transporter TmrAB[13].

As previously reported[29], and in agreement with the cryo-EM structures, DEER measurement of distance distributions for the intracellular spin label pair 96$^{BmrC}$/188$^{BmrD}$ (96/188) shows a predominant IF population under all conditions. A minor distance population corresponding to the putative OF conformation is identified in the high energy post-hydrolysis state (HES) (trapped by vanadate following ATP turnover) but not upon AMP-PNP (a non-hydrolyzable analog of ATP) addition (Fig. 3b, c, Supplementary Fig. 12). Notably, in the QQ background (Fig. 3d), the distance population stabilized by ATP binding, and associated with closing of the intracellular side, is larger compared to the HES in the WT background suggesting a stabilization of an OC conformation by the QQ mutations. Although conformational changes were not detected on the extracellular side in the cryo-EM structures,

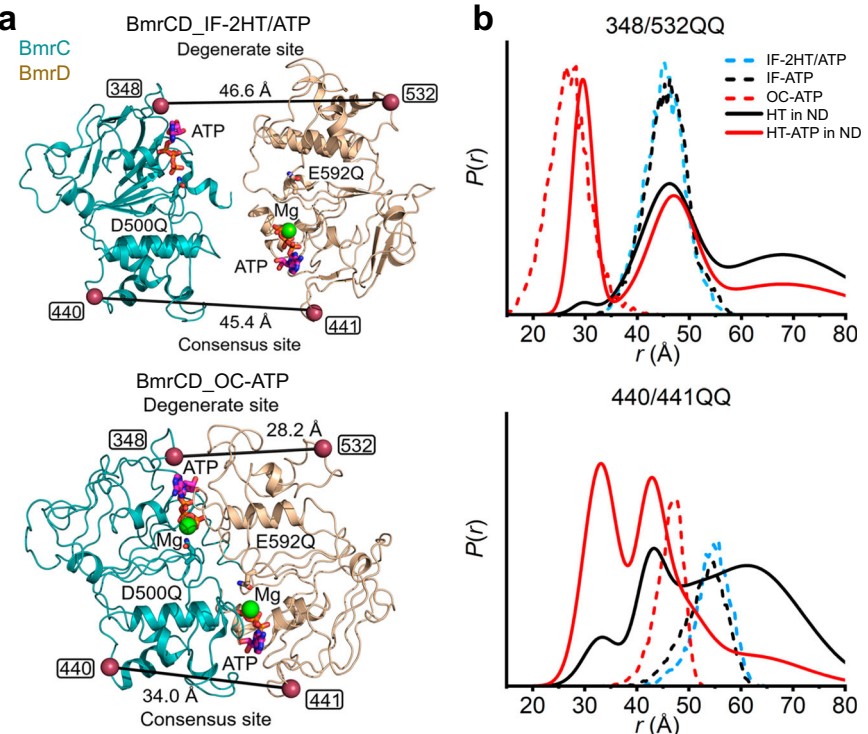

**Fig. 4 | DEER distance distributions for spin-labeled pairs monitoring the NBSs.** **a** Closed-up view of the NBDs of BmrCD cryo-EM structures (IF-2HT/ATP and OC-ATP) showing the spin label pairs in the consensus and degenerate NBSs respectively. The spin label pairs are highlighted by raspberry spheres. The side chain of general catalytic residues and ATP are represented by sticks. **b** Experimental (solid lines) and predicted (dashed lines) distance distributions highlighting the asymmetry of the NBSs in the OC conformation.

the distance distribution of the extracellular pair 55[BmrC]/146[BmrD] (55/146) shows a minor OF population (Fig. 3c, d, Supplementary Fig. 12) at longer distance compared to the predictions from the OC structure. Attempts to stabilize and determine the structure of the OF conformation by cryo-EM were not successful.

## NBD dimer assembly and differential Mg²⁺ binding

Supplementary Fig. 13a shows the details of how BmrCD engages bound ATP in the consensus and degenerate NBSs in the OC conformation. At the degenerate NBS (Supplementary Fig. 13a, b), BmrC residue Y346 (typically part of the conserved A-loop) and S565 of BmrD interacts with the adenine ring of ATP. The phosphate groups are coordinated by residues G374, S375, G376, and K377 from Walker A of BmrC, Q419 from the Q-loop of BmrC, and S568 and G570 of BmrD (Supplementary Fig. 13b).

Binding of the ATP molecule at the consensus NBS shows small but distinct differences relative to the degenerate NBS. In contrast to the equivalent residues at the degenerate NBS, S476 and Q479 of BmrC stabilize the ribose of ATP. G478 on the ABC signature motif of BmrC, G468, K469, S470, and S471 on Walker A and Q592 on Walker B of BmrD, and Q511, H623 of BmrD contribute to phosphate group binding (Supplementary Fig. 13c). Interestingly, comparison of the catalytic residues D500 in the degenerate NBS and E592 in consensus NBS (both mutated to glutamines) in BmrCD_IF-2HT/ATP, BmrCD_IF-ATP, and BmrCD_OC-ATP, reveals that the side chain of Q500 is oriented toward the γ-phosphate group of ATP. On the other hand, the side chain of Q592 flips its orientation toward the γ-phosphate group of ATP in OC while it points away in the IF (Supplementary Fig. 14). This phenomenon was also reported in TmrAB at canonical and non-canonical NBSs, in which the side chain of the former flips but the other does not[13].

Comparison of the NBSs in the three nanodisc structures revealed that one of the IF conformations, obtained in the absence of Hoechst

(BmrCD_IF-ATP), lacks density corresponding to the Mg²⁺ ions, whereas the IF obtained with Hoechst has discernable Mg²⁺ density only in the consensus NBS. The space for Mg²⁺ is occupied by S470 in the consensus NBS and T378 in the degenerate NBS (Supplementary Figs. 1 and 15). Because the quality of the densities in this region for the three structures appears equivalent, we surmise that these observations reflect the affinity for Mg²⁺, being preferentially stabilized at the consensus NBS prior to engaging in ATP hydrolysis.

## Structural asymmetry of the NBSs

Remarkably, despite the catalytic impairment of the BmrCD construct, we observed stark structural asymmetry in the assembly of the two NBSs upon transition from the IF to the OC conformations. To highlight and experimentally support the distinct arrangement of the two NBSs, we compared experimental distance distributions in lipid bilayers to those predicted based on the cryo-EM structures (Fig. 4, Supplementary Fig. 16). For this purpose, distance distributions for spin label pairs monitoring the two NBSs were calculated in all three BmrCD structures (BmrCD_IF-2HT/ATP, BmrCD_IF-ATP, and BmrCD_OC-ATP using CHARMHH-GUI server and the MDDS method[43]). These distance distributions at the two NBSs (348[BmrC]/532[BmrD] in the degenerate NBS (348/532) and 440[BmrC]/441[BmrD] in the consensus NBS (440/441)) of BmrCD_IF-2HT/ATP and BmrCD_IF-ATP overlap. Predicted distances distributions are drastically shorter and manifestly asymmetric in BmrCD_OC-ATP (Fig. 4b red dashed traces). The corresponding distance distributions predicted by the BmrCD_det structure are shorter and symmetric (Supplementary Fig. 16)[12] suggesting that lipids may stabilize structural asymmetry.

Experimental distance distributions obtained in the QQ background in lipid bilayers are multi-component and asymmetric, reflecting the coexistence of multiple conformations. At the degenerate NBS, the distribution obtained in the presence of Hoechst (Fig. 4b) overlaps the one predicted from the IF structures. In contrast,

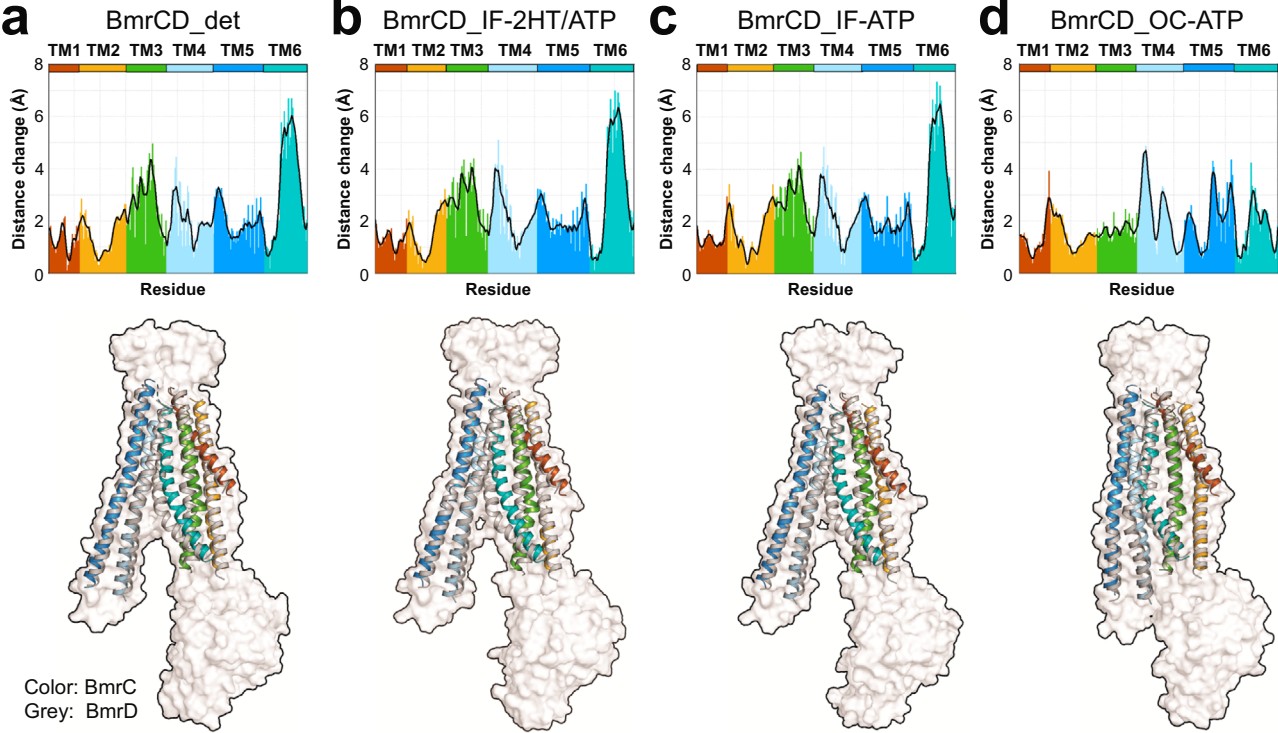

**Fig. 5 | Asymmetric TMDs arrangements underpin BmrCD conformational cycle.** Asymmetric TMDs arrangements analysis for BmrCD_det (**a**) BmrCD_IF-2HT/ATP (**b**) BmrCD_IF-ATP (**c**) BmrCD_OC-ATP (**d**). Upper panels: Distances between alpha carbons in the TMDs of BmrC and BmrD generated by TM-align following superposition of the two TMD. The BmrCD structure in detergent micelles is shown for reference. Lower panels: cartoon representation of the superimposed TM helices from BmrC (colored to match the upper panels) and BmrD (gray). A transparent surface representation of BmrD is shown as reference.

a second major population was observed at ~45 Å at the consensus pair 440/441 in nanodiscs under the same conditions.

Binding of ATP and Hoechst yields an experimental distribution that is almost superimposable to the predicted one from the OC-ATP at the degenerate NBS, although a component at a longer distance overlapping the IF conformation persists. In contrast, at the consensus NBS the predicted and experimental distributions deviate substantially. Two populations are observed in the latter compared to a unimodal population at a longer average distance in the former (Fig. 4). The prominent short component in the experimental distribution is almost superimposable to a similar component for the same consensus pair obtained by vanadate trapping following ATP hydrolysis in the WT background (Supplementary Fig. 16). Notably, this component is stabilized by Hoechst binding suggesting that it corresponds to the OF conformation. Therefore, we conclude that the OC structure does not capture certain structural features at the consensus NBS, consistent with the lack of opening of the extracellular side in this structure noted in Fig. 3.

**Dynamics and asymmetry of the TMD**

In addition to the large ATP-dependent structural changes underpinning the IF to OC transition, we observed changes in the symmetry arrangement of the BmrC and BmrD TMDs. Superposition of the two halves of the TMD around the pseudosymmetry axis in the three nanodiscs and in the previous detergent BmrCD structures (Fig. 5, Supplementary Fig. 17) highlights outward shifts of the TM helices in BmrD relative to BmrC[12]. These outward shifts are observed at both the extracellular and intracellular sides in all structures but to different extents (Supplementary Figs. 17 and 18), suggesting the TM helices on the two sides of the membrane may undergo independent movement.

To quantitatively capture the asymmetry across conformations in lipid bilayers, we measured distance changes between corresponding helices of BmrC and BmrD using TM-align[44] (Fig. 5). For both IF structures, the distance changes reveal increased asymmetry relative to the detergent structure for TM4 on the extracellular side and TM2 and TM4 on the intracellular side (Fig. 5a–c) implying that the lipid environment is more conducive for asymmetry between BmrC and BmrD.

Drastic change in asymmetry is observed upon the transition to the OC conformation. At the extracellular side there is an increase in the asymmetry of TM1 and TM2 that may indicate initiation of transition to an OF conformation even though the extracellular side remains closed in the OC conformation. Remarkably, concomitant movements in TM2, TM3, and TM6 reduce the local asymmetry which we attribute to the loss of interaction with Hoechst (Supplementary Fig. 19). In contrast, TM4's asymmetry is accentuated around a kink in the bilayer.

**The unique ECD domain of BmrCD twists to open**

BmrCD is the only structurally characterized heterodimer that harbors an extracellular domain, the deletion of which abolishes substrate stimulation of ATP hydrolysis[12] (Supplementary Fig. 1). Superposition of ECDs from BmrCD_IF-2HT/ATP, BmrCD_IF-ATP, and BmrCD_OC-ATP, based on BmrC alignment, exposes a twisting motion of this domain in the latter. The anatomy of this twist involves movements of the loop between residues 118 and 127, the loop between residues 73 and 89, and the loop between residues 67 and 70 of 13 Å, 20 Å, and 10 Å, respectively (Fig. 6a). This twist alters the electrostatics of the domain as a whole, displacing a patch of positive charges, which could potentially enable an exit pathway for substrate release as it is squeezed out by the transition from IF to OC. (Fig. 6b). Meanwhile, the charge distribution of the ECD docking surface to the extracellular side of the TMD remains invariable in all structures (Supplementary Fig. 20) implying that the ECD movement is independent of the TMD.

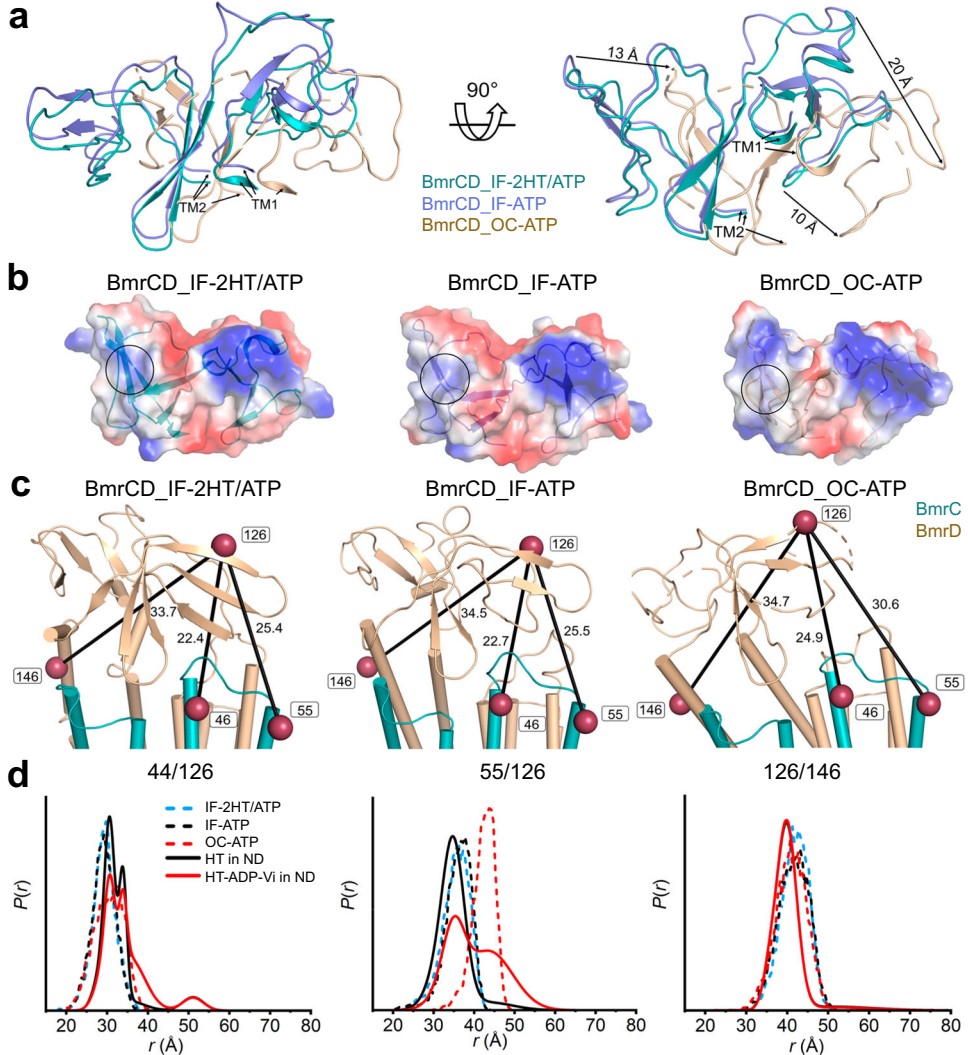

**Fig. 6 | Movement of the ECD opens a putative path for substrate efflux.**
**a** Superimposition of ECD based on the alignment of BmrC in the BmrCD cryo-EM structures highlighting the movement of the loops. **b** Electrostatics potential surface of ECD in BmrCD cryo-EM structures shown from an extracellular view. The color scale ranges from blue (positively charged regions) to red (negatively charged regions) whereas neutral regions are shown as white. The positive charged patch which possibly blocks the substrate exit in the IF structures is highlighted in a black circle, a corresponding patch in BmrCD_OC-ATP with neutral charge is also highlighted with a black circle for comparison. **c** Cartoon representation showing the three spin label pairs between ECD and TMD. The spin label pairs are illustrated by raspberry spheres connected by a straight line. **d** Experimental (solid lines) and predicted (dashed lines) DEER distance distributions highlighting the movement of the ECD.

We tested if the ATP-powered partial opening movement of the ECD is also observed in the WT background. Distance distributions of three pairs $46^{BmrC}/126^{BmrD}$ (46/126), $55^{BmrC}/126^{BmrD}$ (55/126), and $126^{BmrD}/146^{BmrD}$ (126/146), predicted by MDDS based on the IF structures, are in agreement with the DEER data (Fig. 6c, d, Supplementary Fig. 21). Upon transition to the HES, prominent distance changes are observed between TM2 and the ECD (the 55/126 pair) as predicted by the OC conformation. The structures and the DEER data suggest that part of the ECD is anchored to the TMD specifically by its interactions with TM1 and TM2 of BmrD (the 126/146 pair) and interactions with TM1 and/or TM5 of BmrC (Fig. 6c, d, Supplementary Fig. 22). The relatively broader distributions in the HES (55/126) are indicative of increased dynamics of part of this domain which correspond to the relatively poorer cryo-EM maps of ECD in the OC conformation (Supplementary Fig. 6).

**Lipids preferentially bind the IF conformation**
We observed substantial density around the TMDs in all the high-resolution structures. Based on the shape and length of these densities, we modeled phospholipid molecules in the vicinity of the transporter. To assess the stability of the individual lipids, we monitored the minimum distance of each to the transporter surface (Supplementary Fig. 23a, b) during MD simulations (see methods). This analysis identifies those lipids which may be stably bound to the protein and those that may have been captured by cryo-EM simply due to the nanodiscs environment. In the simulation trajectories, we observed variable stability for the cryo-EM-modeled lipids relative to their initial position and conformation (Supplementary Fig. 23). Overall, the IF conformation appears to be more hospitable to bound lipids, including several lipids near the elbow helix of BmrD (Lipids 2 and 14 in IF) as well as other regions of the protein such as the interface between the two arms of the TMDs (Lipids 3 and 9 in IF) (Supplementary Fig. 23c). The OC conformation on the other hand seems to mostly stabilize lipids around the elbow helix.

To further characterize the stability and dynamics of the lipids during the simulations, we binned lipid phosphorous coordinates within 20 Å of the protein for both the upper and lower leaflets. From this dataset, we performed kernel density estimation to obtain a density map (Fig. 7a, b) in which sites of high-density correlate with stable lipid

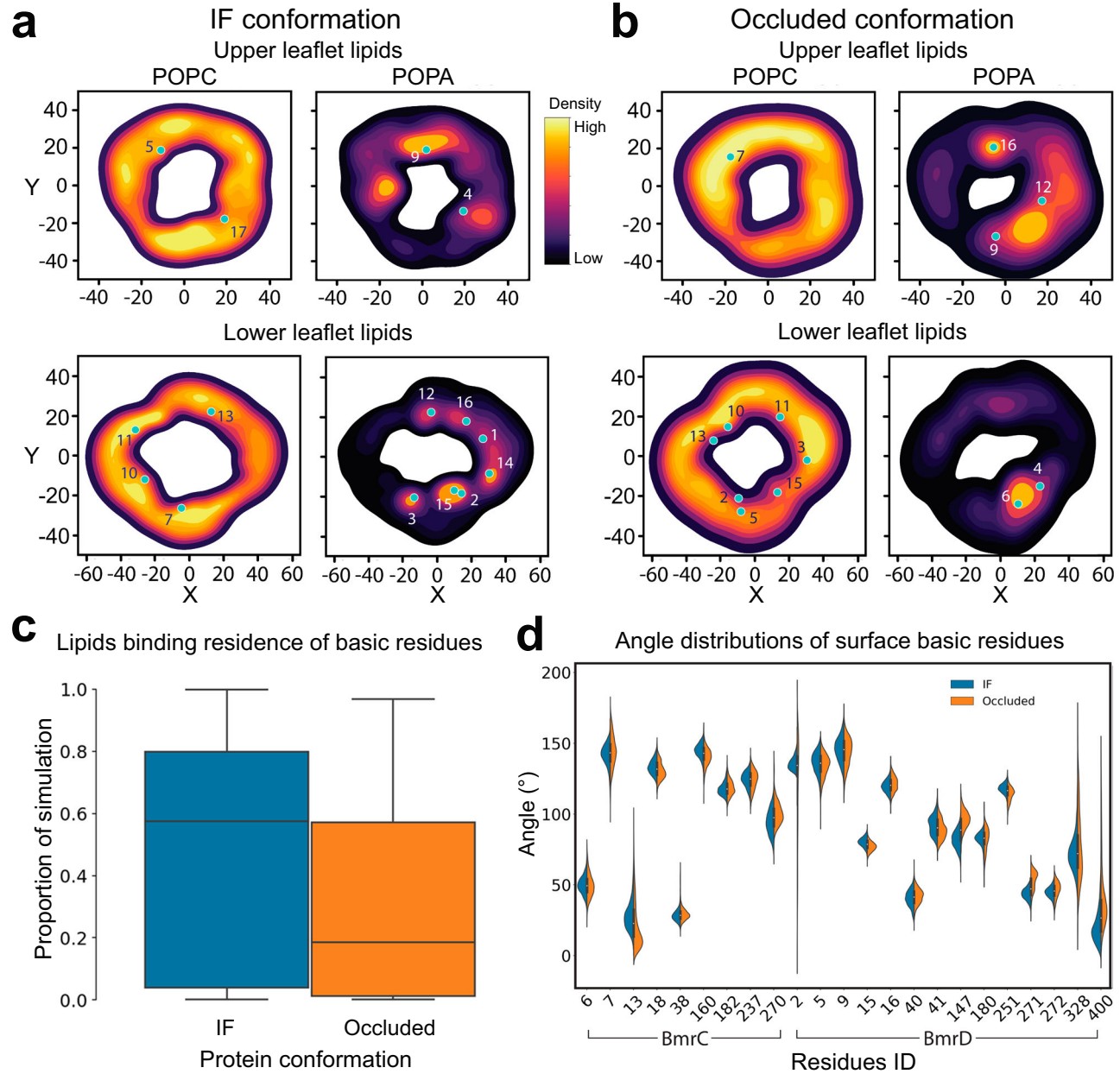

**Fig. 7 | Stably bound lipids identified by MD simulations. a, b** Kernel density estimation performed on lipid phosphorus atoms within 20 Å of BmrCD IF (BmrCD_IF-2HT/ATP) and OC (BmrCD_OC-ATP) conformations. Each leaflet is represented by the X/Y positions in order to map out the density of lipids. Initial position of cryo-EM modeled lipid phosphorus atoms is marked in teal. The regions with oval bright yellow densities highlight stable lipid binding. The regions with smeared densities indicate nearby mobile lipids which do not exhibit stable binding. **c** Protein-lipid residence times for surface basic residues binned into a box-and-whisker plot. Prevalence of stable lipids in the IF conformation is highlighted by larger mean binding residence time. For the IF conformation there were $N = 30$ samples, with minimum value of 0.002, interquartile range of (0.040, 0.800), mean of 0.576, and maximum value of 1.000. For the Occluded conformation there were $N = 33$ samples, with minimum value of 0.002, interquartile range of (0.014, 0.573), mean of 0.187, and maximum value of 0968. **d** Angle space distribution for each surface basic residue for both conformations. Key residues such as K2, R180, and R328 have a tighter distribution in the IF and a broader more flexible distribution for the OC conformation. Details regarding the statistics of each of the 46 distributions are detailed in Supplementary Table 3.

binding whereas smeared densities reflect lipids that are simply gliding along the surface of the protein. For example, it appears that Lipid 13 in the IF conformation is stably bound near its initial placement. However, the density near Lipid 13 is attributed to a different lipid from the bulk bilayer (Supplementary Fig. 23a). Even lipids that seem to unbind in some but not all simulation replicas, such as Lipid 9 (Supplementary Fig. 23a), appear to stay proximal to the protein in the unbound replicas.

For the OC conformation, the kernel densities highlight that Lipids 6 and 16 appear stably bound, being positioned directly in high density wells (Fig. 7b). Lipid 4, which fluctuates away from the protein

in simulations, does occupy the same high-density region as Lipid 6 near the elbow helix region of BmrD (Fig. 7b, Supplementary Fig. 23b and c). While in some replicas Lipid 4 does not stay fully bound, the convergence at the end of the simulation indicates that it binds in this region albeit less strongly than the equivalent lipid (Lipid 2) in the IF conformation (Supplementary Fig. 23). Interestingly, Lipids 10 and 15 are found to be stably bound according to the distance analysis but the density maps show that, in both cases, these lipids are moving along the protein surface and not settled in a high density well (Fig. 7b, Supplementary Fig. 23b)

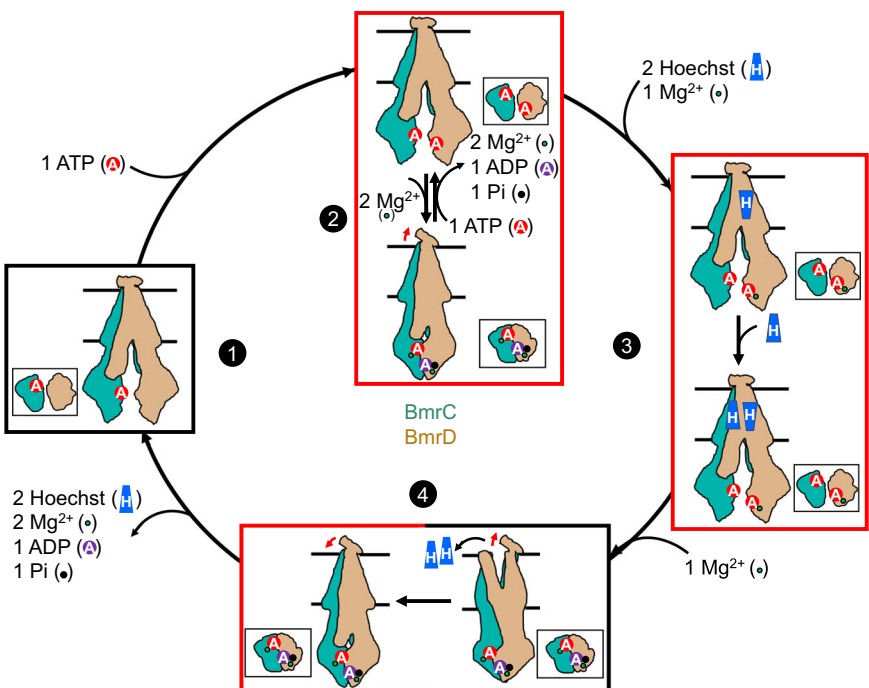

**Fig. 8 | Model of BmrCD transport cycle.** BmrC and BmrD are shown in the same color as Fig. 1. Number refers to the four steps of BmrCD transport cycle: 1→2: ATP binding and the substrate uncoupled cycle, where ATP is hydrolyzed without transport of substrates; 2→3: Hoechst and the first $Mg^{2+}$ binding; 3→4: the second $Mg^{2+}$ binding, ATP hydrolysis and Hoechst release; 4→1: ADP and Pi release which resets the transport cycle. Arrow near the ECD refers the ECD twists to open/close the exit. Red box represents the cryo-EM structures reported here. Black box are conformations deduced from cryo-EM 2D classification or DEER data. ATP, ADP, and substrate Hoechst are represented by red filled A circle, purple filled A circle, and blue filled H trapezoid, respectively. Released phosphate and $Mg^{2+}$ are shown in black and green dot, respectively. The two H trapezoids represent the antiparallel arrangement of Hoechst molecules in the chamber. The top views of the NBDs are shown as insets.

To identify BmrCD's contributions to these protein-lipid interactions, we analyzed the binding residence times of the modeled lipids at basic surface residues (Fig. 7c). These residence times further highlight the propensity for IF to stabilize lipids over the OC conformation. To probe the origin of this propensity, we measured the distributions of side chain angles of basic surface residues in both conformations (Fig. 7d). For this purpose, this angle is defined by two vectors: one from the center of mass (CoM) of the backbone atoms of neighboring residues, and the other vector is drawn from the alpha carbon to the CoM of the terminal charged nitrogen (NZ for lysine and NH1/NH2 for arginine). In cases of stable lipid coordination, the available angle space which can be explored by a given residue is reduced as is observed for several major residues including K2 (BmrD) of the elbow helix in the IF conformation (Supplementary Fig. 23c). Other notable residues are K182 (BmrC), which lies at the vertex of TMD, and R180 (BmrD) which sits along the interface between BmrC and BmrD near the elbow helix.

The apparent importance of the elbow helix in recruiting lipids focused our attention on key residues which might be inducing strong local protein-lipid interactions. Several charged residues, K2, K5, and R15 of BmrD, were in silico mutated to alanine in order to assess the effects of charge interactions on bound-lipid residence time relative to WT (Supplementary Fig. 23d). In all cases and for both conformations, we observed a decrease in the high residence binding regime. This was coupled with a lack of perturbation in the low residence binding as expected for surface residues.

## Discussion
### Conformational dynamics of BmrCD
The integrated investigation of BmrCD presented above addresses two long-standing questions in the field-namely the origin and the role of asymmetry in the conformational cycle of ABC transporters. We present direct experimental evidence of asymmetric substrate and $Mg^{2+}$ binding that appear to be coupled to the TMD and NBD structural asymmetries. Furthermore, our study contributes structural and dynamic insights into how lipid/transporter interactions can modulate function with implications to membrane proteins in general. Three snapshots of the ATP-powered conformational cycle in lipid bilayers describe how ATP binding engenders transition to an OC state in which cytoplasmic access to the chamber at the interface of the two TMDs is closed. The structural elements that enable this transition are conserved among ABC transporters as illustrated in the comparison between corresponding intermediates of BmrCD and TmrAB (Supplementary Figs. 1 and 24). That a population of the ATP-bound conformation is also inward-facing, both in detergent micelles and lipid bilayers, suggests that energetics of BmrCD conformational equilibrium deviate from TmrAB and MsbA, in which ATP binding triggers NBD dimerization and extracellular gate opening[13,14,45]. Finally, the lack of an OF conformation in the QQ background can be ascribed partially to the previously demonstrated requirement for ATP hydrolysis to stabilize this intermediate[29]. However, we were unable to capture this conformation in the WT background under turnover conditions or in the vanadate-trapped HES which may reflect the intrinsic heterogeneity and/or instability of this conformation. This conclusion is consistent with the relatively transient nature of this state inferred from multiple studies of ABC transporters[19,46].

Figure 8 summarizes our BmrCD transport model, updated based on the results presented in this study, that emphasizes the interplay between asymmetry and ligand binding. BmrCD partitions into an ATP-powered but substrate-uncoupled cycle (Step 2). Based on DEER data[29], an OF conformation is not significantly populated in this cycle in lipid bilayers. Similar to Pgp and other ABC exporters, this cycle is characterized by lower ATP turnover rate relative to the substrate-coupled cycle. Subsequent to Hoechst binding, which we surmise

occurs in two distinct steps as evidenced by two IF conformations with one or two bound Hoechst, BmrCD partitions to the ATP- and substrate-coupled cycle. Since BmrCD bound to ATP populates the IF conformation (Step 1) and appears to bind at least one ATP at the degenerate NBS[31,47], we propose that binding of the second ATP followed by sequential coordination by $Mg^{2+}$ represent potential rate limiting steps for triggering ATP hydrolysis (Steps 2, 3 and 4). In support of this contention, we could only identify one $Mg^{2+}$ in the cryo-EM structure of the ATP- and Hoechst-bound IF conformation (Step 3, Supplementary Fig. 15). Finally, despite the absence of the OF from the current structural records, the DEER data unequivocally suggest the population of an OF conformation of BmrCD that has features similar to the corresponding MsbA conformation[29]. Therefore, substrate extrusion out of the chamber is executed at this step. We propose that the OC structure captured here represents a return step from the OF state, since it is likely that the ATP at the consensus NBS is already hydrolyzed (Step 4). Because previous DEER analysis indicated that population of the OF conformation requires ATP hydrolysis[29], we expect that the return step in the WT background to have ADP molecule in the consensus NBS.

### The impact of lipid bilayers on energetics of the conformational cycle of ABC transporters

We have previously shown that reconstitution of BmrCD into lipid bilayers attenuates the coupling between ATP hydrolysis and isomerization to the OF conformation despite a 11-14-fold enhancement of ATP hydrolysis rate[29]. It was speculated that this could reflect a coupling leak wherein the dimerization of the NBDs is not efficiently coupled to the TMD. This result was distinct from the coupling observed in TM287/288[48] and TmrAB[13]. The mechanism responsible for the loss of efficiency in BmrCD, however, was not described.

MD simulation has provided important insight into the dynamics of ABC transporters, particularly heterodimers[48–50]. The combination of cryo-EM visualization of lipids and MD simulations to triage non-specifically bound lipids enables the formulation of a plausible hypothesis to explain this observation. More lipid molecules stably bind to the IF conformation relative to OC. Importantly, we observed stably-bound lipids at the interface between the two TMD leaflets that would have to be displaced for the transition to OC or OF conformations. For example, Lipids 3 and 9 bind to the interface of the TMDs in the IF conformation within 400 ns and 300 ns in the MD simulations, respectively, whereas just one lipid (Lipid 16) is stably bound at the interface in the OC conformation (Supplementary Fig. 23a–c). A wide IF conformation has been reported for multiple ABC transporters[12,13,16,31,51–54] which could imply that lipid interactions and consequent modulation of the energetics of conformational changes may be a ubiquitous element for these transporters. While a more detailed analysis of the role of lipids, including the headgroups and the chain length, in the mechanism of ABC exporters is yet to be fully investigated, the results presented here suggest a critical role for lipid-transporter interactions.

### Structural asymmetry in the transport cycle of ABC exporters

ABC heterodimers with a catalytically impaired NBS has presented challenges to existing models derived from investigations of canonical ABC exporters with two active NBSs. Specifically, it raised questions regarding the role of the impaired NBS in priming the transporter and how asymmetry is propagated between the TMD and NBD. The findings reported here may have more general implications to type IV ABC exporters and possibly across ABC exporters. Our previous DEER investigation of the ATP- and substrate-coupled conformational cycle of the mammalian ABC exporter Pgp, which were partly the impetus of the systematic study of BmrCD, revealed that asymmetric conformations at the NBSs are differentially stabilized by substrate and inhibitors suggesting a critical role of these conformations in the transport

cycle[26,27]. Whereas BmrCD has intrinsically catalytically inequivalent NBSs, we found that structural asymmetry, stabilized by substrate binding in the TMD, propagates to the NBSs where it results in exclusive ATP hydrolysis at the consensus NBS. We surmise that differential interactions of ATP and $Mg^{2+}$ with the NBSs, as visualized in our structures (Supplementary Fig. 13), provide a basis for the catalytic inequivalence. Despite the QQ background, it is remarkable that only the side chain of the consensus catalytic residue flips in order to coordinate the nucleophilic attack on the ATP γ-phosphate. Thus, the cryo-EM structures presented here build the notion that such asymmetry may be a conserved aspect of ABC exporter transport cycles. Further investigations of ABC exporters in lipid bilayers without functionally impairing mutations will be required to further illuminate this aspect of the transport mechanism.

## Methods

### Cloning, expression, and purification of BmrCD

Wild type BmrCD in pET21b(+) was used as the original template to generate the cysteine-less BmrCD (BmrCD*) and D500Q in BmrC, E592Q in BmrD contained BmrCD* (BmrCD*-QQ) as described previously[12,29]. Briefly, three native cysteines, C154, C256, and C351, in BmrD were substituted with alanine using site-directed mutagenesis to create cysteine-less BmrCD. To impair the ATP turnover of BmrCD, the catalytic residues D500 in BmrC and E592 in BmrD were mutated into glutamine by site-directed mutagenesis.

WT BmrCD and BmrCD*-QQ plasmids were transformed into *Escherichia coli* BL21(DE3) cells and the transporter was expressed in minimal media supplemented with glycerol (0.5% v/v), thiamin (2.5 μg/ml), ampicillin (100 μg/ml), MgSO4 (1 mM) and 50× MEM amino acids. Cell cultures were grown at 37 °C until optical density of OD600 reached ~1.2, and then the temperature was decreased to 25 °C with 0.7 isopropyl β-d-1-thiogalactopyranoside (IPTG) added to induce BmrCD expression. After 5 h expression, the cells were harvested and frozen at −80 °C. Cell pellets were resuspended in lysis buffer (50 mM Tris-HCl, pH 8.0, 1 mM EDTA, 1 mM PMSF, and a complete EDTA-free protease inhibitor cocktail tablet (Roche)). After sonication, the lysate was centrifuged at 9000 x *g* for 10 min at 4 °C, and supernatant was used for another round of centrifugation at 185,000 x *g* for 1 h at 4 °C to get the membrane pellets. The membrane pellets were solubilized in resuspension buffer (50 mM Tris-HCl, pH 8.0, 0.1 M NaCl, 15% (v/v) glycerol, 1 mM DTT and 1.25% w/v n-dodecyl-β-D-maltopyranoside (β-DDM) for 1 h on ice). The sample was centrifugated at 185,000 x *g* for 1 h at 4 °C, and the supernatant was mixed with the pre-equilibrated Ni-NTA beads for about 2 h in binding buffer (50 mM Tris-HCl, pH 8.0, 0.1 M NaCl, 15% (v/v) glycerol, 0.05% β-DDM). The binding buffer supplemented with 20 mM and 250 mM imidazole were used for resin wash and protein elution, respectively. Eluted fractions containing BmrCD were further purified by size exclusion chromatography (SEC) using a Superdex 200 Increase 10/300 column (Cytiva) for further purification in the SEC buffer 1 (50 mM Tris-HCl, pH 7.5, 0.15 M NaCl, 10% (v/v) glycerol, 0.05% β-DDM).

### Nanodiscs preparation of BmrCD

The purified MSP and BmrCD in detergent were reconstituted into nanodiscs as described previously[29]. Briefly, two lipid mixtures were made. PC (L-α phosphatidylcholine) (Cat #: 840051, Avanti Polar Lipids, Alabaster, USA) and PA (L-α phosphatidic acid) (Cat #: 840101, Avanti Polar Lipids, Alabaster, USA) were mixed as a 9:1 molar ratio or PC and *E. coli* polar lipid extract (Cat #: 100600, Avanti Polar Lipids, Alabaster, USA) were combined in a 1:3 molar ratio. Lipids were then mixed with MSP and BmrCD micelles at a molar ratio of 1800:360:3:1 (β-DDM: lipid: MSP: BmrCD). The mixtures were rotated for 30 min, and then biobeads (0.8–1 g/ml) were added to rotate overnight at 4 °C to remove the detergent. The biobeads were precipitated by centrifugation at 600 x *g* for 5 min and then removed completely using a

0.45 μm filter. BmrCD in nanodiscs was separated by size exclusion chromatography using a Superdex 200 Increase 10/300 column (Cytiva) with SEC buffer 2 composed of 50 mM Tris-HCl, pH 8.0, 0.15 M NaCl (For cryo-EM) or SEC buffer 3 composed of 50 mM Tris-HCl, pH 7.5, 0.15 M NaCl, 10% glycerol (For CW-EPR and DEER).

## Cryo-EM sample preparation and data acquisition

Reconstituted BmrCD*-QQ (~3 mg/ml) was incubated on ice for 30 min after supplementing with 10 mM ATP, 5 mM $MgCl_2$, and with or without 1.2 mM Hoechst-33342 (Thermo Fisher Scientific). To prepare vanadate trapped BmrCD sample, reconstituted wild type BmrCD (~3 mg/ml) was incubated at 37 °C for 15 min with 10 mM ATP, 4 mM $MgSO_4$, and 4 mM vanadate before plunging. For cryo-EM grids preparation, 2.5 μl of ~3 mg/ml BmrCD was placed onto the Quantifoil UltrAuFoil grids (R1.2/1.3, 300 mesh, Electron Microscopy Sciences) which were glow discharged for 20 sec at 25 mA. Then the grids were blotted for 3 sec at force 12 using two layers of filter papers and were plunge-frozen in liquid ethane by Vitrobot Mark IV (Thermo Fisher) with the environmental chamber set to 4 °C and 100% humidity. Cryo-EM grids for data collection were selected after screening by TF20 or Glacios (Thermo Fisher Scientific).

Cryo-EM data of BmrCD_IF-2HT/ATP were collected by Titan Krios G4 microscope (Thermo Fisher Scientific) at 300 keV, equipped with Gatan BioQuantum filter, Gatan K3 with GIF detector. Images were recorded with two exposures per hole using EPU in super-resolution mode with a 20 eV slit width of energy filter and at a magnification of 105,000 with a pixel size of 0.818 Å/pixel. Defocus was set to vary from −0.8 to −2.2 μm. Each image was dose fractionated to 50 frames. The total dose over the sample was 48.271 $e^-/Å^2$ and the dose was 0.965 $e^-/Å^2$/frame. 5,696 movies were collected for this data set. Cryo-EM data of BmrCD_OC-ATP was collected using a similar condition except with a higher magnification of 130,000 with a pixel size of 0.647 Å/pixel. The entire dataset contains 11,829 movies.

Cryo-EM data of BmrCD_IF-ATP was imaged with a 300 keV FEI Krios G3i microscope equipped with a Gatan K3 direct electron. Movies containing 40–50 frames were collected at a magnification of 81,000 in super-resolution mode with a pixel size of 0.55 Å/pixel and defocus range of −0.8 to −1.6 μm using automated imaging software EPU (Thermo Fisher Scientific).

## Cryo-EM data processing

All datasets were processed in RELION 3.0[55] and cryoSPARC[42]. MotionCor2 and Gtcf were used to perform beam-induced motion correction and CTF estimations, respectively. Images with poor quality were discarded.

For BmrCD_IF-2HT/ATP dataset processing, auto picking implemented with Laplacian-of-Gaussian method of Relion was used to pick around 3 million particles, and extracted as 4 × 4 binned, through 6 rounds of 2D classification and one round of 3D classification using rescaled BmrCD map (EMD-23641)[12]. After 4 more rounds of 2D classification, the particles were unbinned and one more 3D classification was carried out to obtain a 4 Å map. Using this map as a template, we did template-based picking, multiple rounds of 2D classification, and 2 rounds of 3D classification to obtain a class with 133,280 particles. Then processing was implemented with cryoSPARC 3.2.0[42], performing NU-refinement and local refinement to yield a final map at 3.34 Å. To improve the quality of the map, more rounds of local refinements were applied using masks covering parts of the original map. Four separate masks that cover ECD, TMD, BmrC-NBD, and BmrD-NBD were used (Supplementary Fig. 4).

For BmrCD_IF-ATP dataset processing, a similar processing procedure was performed. Briefly, after several rounds of 2D classification and 3D classification, a class containing 529,549 particles was transferred to cryoSPARC 3.2.0[42] to do Ab-initio, then a class was chosen to

run NU-refinement and local refinement. Finally, a cryo-EM map with 3.27 Å resolution was obtained. More rounds of local refinements for masks on parts of the original map were also applied to improve the quality of the density.

BmrCD_OC-ATP dataset processing was similar to BmrCD_IF-ATP dataset processing except that this dataset displayed anisotropy. To solve the anisotropy of the map, we performed heterogeneous refinement. Then NU-refinement and local refinement were performed to get the final map at 2.9 Å. More rounds of local refinements for masks on ECD, TMD and closed NBD of the original map were also applied to improve the quality of the density. A similar processing procedure is applied to obtain the map of BmrCD_IF-ATP2 in this dataset.

Dataset 2 and dataset 5 in Supplementary Fig. 2 were processed using the same strategies as above (Supplementary Fig. 3).

## Model building and figure preparation

The same pipeline for model building process was performed for all the structures. The initial model was searched with dock_in_map implemented in Phenix by separated domains from the BmrCD structure in detergent micelles (PDB ID: 7m33)[12]. Combined with manual adjustment in coot[56] and refinement by real-space refinement implemented in Phenix[57], a satisfactory model was built. MolProbity implemented in Phenix[58] was used for all the structural model validation. For BmrCD_IF-HT/ADPVi in Supplementary Fig. 2, the model of BmrCD_IF-2HT/ATP was fitted to the density in Chimera manually due to its low resolution.

Figures were prepared by UCSF Chimera[59], ChimeraX[60], and PyMol. Solvent-accessible volume calculated by CASTp[40].

## Molecular dynamics simulations

**Equilibrium simulations.** Initial placement of BmrCD in the membrane and arrangement of lipids around it were done by the CHARMM-GUI webserver[43,61] using the published structure (PDB ID: 7M33[12]) as a proxy for our Cryo-EM models. To match the cryo-EM experimental conditions and the nanodisc environment in which our structures were determined, we chose to model the protein in a symmetric membrane consisting of 9:1 POPC:POPA. Partially resolved cryo-EM lipids were classified as POPC or POPA depending on the resolved head group atoms. Lipids with resolved Nitrogen and Phosphate atoms were labeled as POPC, and those with resolved P atoms only were labeled as POPA. Lipid tails without any headgroup atoms were excluded from the modeling process. Missing atoms in the cryo-EM lipids were completed using the PSFGEN plugin in VMD[62]. Missing atoms of the protein were completed with MOE (Chemical Computing Group, Montreal, Canada). The protonation states of the residues were assigned using PropKa[63]. Completed cryo-EM lipids were combined with the protein, Hoechst-33342 and two ATP molecules preserving their position in the cryo-EM structure. Upon visual inspection, any lipid tails modeled artefactually penetrating the protein core were rotated around one of the C-C bonds to achieve a better initial position and orientation without changing the positions of the cryo-EM resolved atoms. The protein structure was placed into a bilayer using the CHARMM-GUI web server and cryo-EM lipids were retained in their position[43,61]. Any bulk membrane lipids within 0.5 Å of either the protein or the completed cryo-EM lipids were removed to prevent clashing. This model was then re-ionized to ensure charge neutrality using the AUTOIONIZE plugin in VMD. Parameters for the Hoechst-33342 ligand present in the open conformation are previously reported in our past work[12]. Replica simulations were generated using the Membrane Mixer tool in VMD[64] to better sample the initial lipid distribution for the non-modeled lipids.

All the simulations mentioned previously were conducted using NAMD2.14[65,66], incorporating CHARMM36m[67] and CHARMM36 force

fields for the proteins and lipids[68,69], respectively, with the following parameters utilized by default unless stated otherwise. The TIP3P water model[70] was employed. Non-bonded interactions were computed with a 12 Å cutoff and a switching distance of 10 Å. To estimate long-range electrostatic interactions, the particle mesh Ewald (PME) method was utilized[71]. A Langevin thermostat with a damping coefficient of 1.0 ps^(−1) was employed to maintain a constant temperature of 310 K. The Nosé–Hoover Langevin piston method was used to uphold the pressure at 1 bar[72]. A flexible cell was enabled to allow motion along all three dimensions independently, while the $x/y$ ratio of the membrane remained constant. Bonds involving hydrogen atoms were maintained as rigid using SHAKE[73] and SETTLE algorithms[74]. A time step of 2 fs was used for all equilibrium simulations and production runs.

Each simulation system was energy-minimized for 10,000 steps followed by a gentle heating scheme of incrementing the system temperature by 25 K for 100 ps at a time until the target temperature of 310 K was achieved. Equilibration was performed using NAMD2.14[65,66] for 7.5 ns with restraints following this scheme: 2.5 ns with protein backbone and lipid headgroups restrained at 1 kcal/mol/Å², 2.5 ns with just protein backbone restrained at 0.5 kcal/mol/Å² and a final 2.5 ns equilibration with protein $\alpha$-carbons restrained at 0.5 kcal/mol/Å² to allow the lipid tails to melt and the protein side chains to relax. Restrained equilibration was then followed by 5 ns of simulation without restraints during which pressure coupling was turned on. To allow modeled lipid tails to blend into the membrane, a grid-based potential was applied using the Grid Forces[75] option of NAMD during the first phase of equilibration. Production MD was then carried out for 500 ns in each replica for a total of 3 μs of sampling across both conformations and all 3 replicas.

All analysis was performed in VMD and the Seaborn and SciPy python packages[76,77].

**In silico mutagenesis.** K2, K5 and R15 were sites of the BmrD elbow helix identified as participating in highly resident lipid-binding. Each site was individually mutated to an alanine using the Mutagenesis tool in VMD and system charge once again corrected using the AUTO-IONIZE plugin. Three replicas were generated for each mutant for a total of 18 simulations (3 replicas x 3 mutants x 2 conformations). Each replica was treated as before with a slow equilibration and 500 ns of production MD. All analysis was performed in VMD and plotted in the Seaborn python package[77].

The information for the convergence of simulations and details about the MD simulation system setups is shown in as MD Supplementary Data 1. The PDB and PSF files for the initial and final structures in all simulations, as well as sample configuration files for equilibrium and production, and the parameter files used in both IF and OCC MD simulations, are available in MD Supplementary Data 2 and MD Supplementary Data 3. Please note that due to their substantial file size, the trajectory files have not been included in the deposit, but we can share them upon request.

**DEER sample preparation and DEER spectroscopy.** To prepare samples for double electron-electron resonance (DEER), BmrCD double-cysteine mutants eluted from Ni-NTA affinity purification were labeled with 20-fold molar excess of (1-Oxyl-2,2,5,5-tetra-methylpyrroline-3-methyl)methanethiosulfonate (MTSSL, Enzo Life Science) at room temperature in dark for 2 h Followed by incubation with another 20-fold molar excess of MTSSL at room temperature in dark for 2 h and then the sample was moved to 4 °C overnight. The labeled BmrCD samples separated from the free label by Superdex 200 Increase 10/300 column (Cytiva) in SEC buffer 1 for DEER in detergent buffer. Reconstitution and a size-exclusion chromatography in SEC buffer 3 were applied to prepare nanodisc samples for DEER.

Spin-labeled BmrCD mutants were incubated with nucleotides and/or Hoechst-33342 (Thermo Fisher) for DEER. Different combinations were used to capture different states of BmrCD with final concentrations of ATP, vanadate, AMP-PNP, Hoechst-33342, and MgSO₄ at 10 mM, 5 mM, 10 mM, 1 mM, 10 mM, respectively. For high-energy post-hydrolysis state, BmrCD in detergent micelles and in nanodiscs were incubated with ATP, vanadate and MgCl₂ at 30 °C for 30 min and 37 °C for 30 min, respectively. To trap an ATP-bound state, BmrCD in detergent micelles and in nanodiscs were incubated with AMP-PNP and MgSO₄ at 30 °C for 30 min and 37 °C for 30 min, respectively.

For DEER spectroscopy, a final concentration of 24% (v/v) glycerol was used as a cryoprotectant for all DEER experiments. An Elexsys E580 pulsed EPR spectrometer operating at Q-band frequency (33.9 GHz) with a 40 W Amp-Q amplifier (Bruker) with the dead-time free four-pulse sequence at 83 K was used[78,79]. We used the pulse lengths of 10 ns (p/2) and 20 ns (p) for the probe pulses, and 40 ns for the pump pulse, separately. And the frequency separation was set as 73 MHz. Raw DEER decays were analyzed as described previously[12,29]. Briefly, a home-written software operating in the Matlab environment was used to analyze the raw DEER as previously described[80–82].

For distance distributions for the same spin-labeled position under different conditions, global analysis of the DEER decays was applied. Global analysis of DEER decays under different conditions for the same spin label pair was carried out with home-built software as described[29] with a slight modification to the statistical criterion. As before the distance distribution is assumed to consist of a sum of Gaussians. The center and width of the Gaussians were shared across conditions with the amplitudes allowed to vary between conditions. The depth of modulation and the slope of the background were allowed to vary between conditions. Different than the earlier version of the software, the optimal number of Gaussians was determined with the Bayesian information criterion (BIC)[82].

Distance distribution predictions based on BmrCD cryo-EM structures were carried out by the DEER Spin-Pair Distributor at the CHARMM-GUI website[43,83] with default parameters in 10 ns MD simulations with dummy spin labels that mimics the dynamics of MTS spin-label.

### Reporting summary

Further information on research design is available in the Nature Portfolio Reporting Summary linked to this article.

## Data availability

The data that support this study are available from the corresponding authors upon request. The cryo-EM maps of the heterodimeric ABC transporter BmrCD in nanodiscs have been deposited in the Electron Microscopy Data Bank (EMDB) under accession codes EMD-29297 (BmrCD_IF-2HT/ATP); EMD-40908 (BmrCD_IF-1HT/ATP); EMD-29362 (BmrCD_IF-ATP); EMD-41004 (BmrCD_IF-ATP2); EMD-29087 (BmrCD_OC-ATP); EMD-40974 (BmrCD_OC-ADPVi); EMD-41058 [https://www.ebi.ac.uk/pdbe/entry/emdb/EMD−41085] (BmrCD_IF-H/ADPVi); The atomic coordinates have been deposited in the Protein Data Bank (PDB) under accession codes 8FMV (BmrCD_IF-2HT/ATP); 8SZC (BmrCD_IF-1HT/ATP); 8FPF (BmrCD_IF-ATP); 8T3K (BmrCD_IF-ATP2); 8FHK (BmrCD_OC-ATP); 8T1P (BmrCD_OC-ADPVi). For MD simulation, initial coordinates, simulation input files and coordinate files of the final output are provided as Supplementary Data 2 and Supplementary Data 3.

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

## Acknowledgements

This work was supported by NIH grant GM128087 to H.S.M. EM data collections were conducted at the Center for Structural Biology Cryo-EM Facility at Vanderbilt University. We acknowledge the use of the Glacios cryo-TEM, which was acquired by NIH grant S10 OD030292-01. EM data collection for BmrCD_IF-ATP was conducted at Case Western Reserve University by Dr. Kunpeng Li. Molecular simulations were performed on resources provided by the XSEDE grant MCA06N060 (E.T.), and supported by NIH grants P41-GM104601 (E.T.), R24-145965 (E.T.), and R01-GM123455 (E.T.).

## Author contributions

Q.T. performed all the experiments. M.S. and H.S.H. performed the MD simulation under the supervision of E.T. E.K. provided advice for structure determination and analysis. R.A.S. helped with DEER data collection, Fig. 5 preparation and discussion. Q.T. and H.S.M. designed the research, analyzed, and interpreted the structures and wrote the manuscript with input from all authors.

## Competing interests

The authors declare no competing interests.
