## [Peer Review File · Nature Communications]

Asymmetric conformations and lipid interactions shape the ATP-coupled cycle of a heterodimeric ABC transporterReviewers' Comments:

Reviewer #1:

Remarks to the Author:

In this work, the authors investigated a unique heterodimeric ABC transporter, BmrCD, using cryo-EM, DEER, and MD simulation analyses. The findings are considered to be very interesting and worthy of publication, especially the twist of ECD and the bindings of two Hoechst and Mg²⁺. However, the following revisions should be considered before publication.

(1) In the context of the Introduction, the third paragraph (l.64-74) mainly refers to Pgp with two consensus sites as an example. These explanations are a little confusing. The homodimer example is a good comparison to the next paragraph on heterodimers, but even when referring to Pgp, it is necessary to clearly state that it is a transporter consisting of a single chain with two consensus sites. As a detail, ref 22 is first cited as an asymmetric example in l.68-69, but is cited as "asymmetry was not observed" in l.72-74. These are sources of confusion and require more appropriate citations and descriptions.

(2) Regarding ECD, there is no description at all before line 356 (on the second half of the results) [except for the abstract]. Although the authors may have intended to describe ABC transporters in general, the existence of ECD may be characteristic among heterodimeric transporters. The existence of ECD should be suggested at the end of the Introduction and earlier in the Results as a reason why BmrCD prefers the IF conformation (such as near l.197). The reader's understanding will change depending on whether or not they keep in mind the presence or absence of ECD.

(3) One of the most difficult aspects of the authors' discussion of the transport cycle is that ATP hydrolysis appears to occur before NBD dimer formation. l.104 seems a little strange. Considering l.455, l.104 should be modified as follows:
"demonstrating the requirement of ATP hydrolysis to stabilize the conformational intermediate, leading to the OF conformation".

(4) Related to comment 3 above, Fig. 8 of the transportation cycle may be misleading. It may be helpful to draw the top views of the NBDs on the right side (as a similar cycle) or as insets (in each). Although ATP is still bound to both sites in step 3, one is converted to ADP in step 4. It may be desirable to move the bottom figure in step 2 to step 3 (or 4). Although it has not yet been confirmed to what extent the results match the present results, the following cycle can be assumed.

In the heterodimer, one site remains ATP-bound (s step 1). When ATP binds to the other site, it becomes unstable and begins to fluctuate (Step 2). Then, binding the two substrates further induces NBD dimerization (step 3). Dimerization is completed by binding of Mg²⁺ to the degenerate site, hydrolysis is induced at the consensus site, and the ECD is opened, leading to the transition to OF (step 4). Finally, the two substrates are extruded, and ADP and Mg²⁺ are also dissociated, returning to step 1 and starting the next cycle.

The above would correspond to an addition of the "role of ECD" in the heterodimer transporter cycle, which can be called the transport cycle of "ECD-coupled" heterodimeric ABC transporters. In heterodimeric type transporters such as Tm287/288 and CFTR, it is believed that once ATP binds, the NBD dimer moves such that one NBS1 remains tethered (or closed) while the other NBS2 opens and closes (It may be a special case of the constant contact model proposed by Jones and George). This would be due to the fact that one ATP molecule remains bound without being hydrolyzed at the degenerate site (NBS1) (although there may be criticism). However, in BmrCD, due to the presence of ECD (its closure), not only NBS2 but also his NBS1 exhibits a very large opening behavior. It would be helpful if this sort of thing could be shown more clearly (for general readers).

(5) From Figure 2, HT-2 is packed between W290 and H338 (HT-2 should be written in the text). Then, what kind of amino acid interactions stabilize HT-1, bound to both the BmrCD_IF-1HT/ATP and

BmrCD_IF-1HT/ATP structures?

(6) Please add sequence alignments of heterodimeric transporters such as BmrCD, TmrAB, and TM287/288 to Supplementary Materials. Then, please explain the presence or absence of ECD and so on in the Introduction, Discussion, or other.

(7) These are somewhat general questions.

(i) Is it possible to observe a state in which 1ATP and 1ADP are bound, such as when ADP and an ATP analog with low affinity are used?

(ii) To understand the general heterodimer cycle, it is possible to compare the wild type and a mutant lacking the ECD of BmrD. Are the authors considering such an approach?

Then, if it were possible, would the authors imagine it would be a cycle that favors IF as well?

[minor]

(8) l.35: "heterodimeric" ABC transporters & please remove "in general"

(9) l.48-50: The first sentence of the introduction cites only a somewhat older reviews (refs 1-3). It is recommended that a recent review on humans be cited here:

Alam & Locher, *Annu Rev Biophys* 52, 275-300 (2023).

(10) l.78: the catalytic glutamates "(of Walker B motif)" [as information]

(11) l.80: "heterodimeric" ABC exporters

(12) l.100 and through the manuscript: Hoechst or HT

l.149: BmrCD_IF-2HT/ATP

l.161: BmrCD_IF-1HT/ATP (more specific)

l.216: HT-1 & Fig.2: HT-1 and HT-2 (avoiding confusion with 1HT)

(Of course, it is also good to express only H, such as 2H, but we think it is easier to understand if it is written specifically, such as 1H.)

(13) Not only the consensus site but also the degenerate site D500 of BmrC was mutated to Q. What would be the outcome if this mutation is absent? Then, please tell us the authors' intentions or the meaning of both mutations.

(14) Although this study deals with structural transitions and substrate transport of heterodimers and performs computational studies, several recent articles and reviews are not cited. For example, we recommend citing the followings and discussing them in Discussion: Göddeke et al., *JACS* 140, 4543-4551 (2018), Göddeke & Schäfer, *JACS* 142, 12791-12801 (2020), Furuta, *Biochem Soc Trans* 49, 405-414 (2021).

(15) Regarding the transport cycle, in addition to ref 32 (Stockner et al), the following recent article may also be informative: Jones & George, *Int J Mol Sci* 24, 2624 (2023).

Finally, this study appears to be a very sophisticated study of a heterodimer with an ECD. This time (due to time constraints), above, I have mainly made some fairly general comments (please forgive me if I misunderstood the authors' excellent work). Next time, I may comment on the details of the representation of data, descriptions, etc., but I hope that the revised version will be easier to understand and provide deeper insights for many readers.

Reviewer #2:

Remarks to the Author:

In this manuscript Tang, Q. et al. provide an extensive cryo-EM, DEER spectroscopy, and molecular dynamics based investigation on the structure and conformational changes of the ABC transporter BmrCD from *Bacillus subtilis*. Despite a wealth of recent investigations into ABC transporter structure and dynamics, key questions remain regarding the overall conformational cycle and energetic coupling mechanisms of these transporters. Questions about the asymmetric nature of ATP binding/hydrolysis persist across the field for various transporters, and especially for transporters such as BmrCD in which one of the two potential ATP binding sites is degenerate and deficient for ATP hydrolysis. Overall, the analysis of BmrCD presented here by Tang, Q. et al. sheds additional light on the nature of asymmetric ATP binding/hydrolysis in such transporters, and reveals additional details on the coupling of substrate loading in the TMDs with ATP/Mg binding and conformational transitions in the NBDs. The cryo-EM, DEER, and molecular dynamics experiments all appear to be well performed. In its current state the manuscript is quite dense with a large amount of data obtained for different BmrCD constructs under various conditions. While the density and breadth of the manuscript will make it quite difficult for the non-expert to follow, I also believe that such depth and complexity is required to support the authors claims. Moreover, the data presented in this manuscript is timely and highly relevant for all researchers interested in ABC transporter structure and function. For this reason, I only have a few comments to be addressed prior to recommending publication...

1. In this reviewer's opinion, one of the major key findings in this manuscript lies in the differential magnesium binding in the different cryo-EM structures obtained. Such a finding would mark a key aspect of BmrCD function, and would illuminate an essential component to the asymmetric nature of this transporter. However, from the figures provided (Suppl. Fig. 14) it is not apparent to me that the cryo-EM maps are of sufficient resolution and quality to support the presence or absence of magnesium. Placement of ions into cryo-EM maps of small membrane transporters is no trivial task, and I feel that the authors should better support the claims of the presence or absence of magnesium with additional figures (indicating the map threshold values) or supplemental videos that support the modelling or absence of magnesium in their final atomic models. Alternatively, the authors could provide the cryo-EM maps and atomic models to the reviewers for inspection.

2. Along the lines of point 1 above, it is often noted in textbooks and throughout the literature that magnesium is an essential cofactor for ATP, and that "most" ATP within a cell exists pre-complexed with magnesium before Mg-ATP binds to enzymes, proteins, etc... However, in figure 8 all steps of the proposed conformational cycle show ATP binding to BmrCD before magnesium. It seems highly unlikely that such a scenario would occur in a cellular context.

3. In Figure 4 for spin label pairs monitoring the distance between NBS's, for the 440/441 pair it appears that the simulated distance distributions are $\sim 10\text{\AA}$ longer than what would be predicted based on the C-alpha positions shown in panel A. This discrepancy seems too large to be accounted for simply by spin label rotamer populations. Moreover, when incubated with H-ATP in nanodiscs (solid red trace, bottom right panel) the distance distribution for the 440/441 pair seems to accurately reflect the approximate distances indicated for a $\sim 50/50$ mixture of the IF ($\sim 45\text{\AA}$)/OC ($\sim 34\text{\AA}$) conformations observed in the cryo-EM structures. Why do the predicted distributions for the 440/441 pair seem so much longer than what is observed in the cryo-EM structures and in the experimental distribution in the presence of H-ATP?

4. In figure 7a and 7b it is not clear to me why in each leaflet for each conformation there are two kernel density maps? As an example, for the top left of figure 7a for the IF conformation on the upper leaflet, what is the difference between these two kernel density maps? This should be clearly indicated in the figure legend. Also, the figure/legend should be clearly amended to indicate to the reader that these kernel density maps are a view along the Z-axis perpendicular to the membrane, and the axes indicate X-Y direction in angstrom.

5. In the methods section on MD, it is only mentioned that BmrCD was placed into a lipid bilayer using

CHARMM-GUI. What was the lipid composition of the bilayer? Was the bilayer asymmetric or symmetric? Although POPC and POPA were used in the nanodisc reconstitution, based on literature reports of *Bacillus subtilis* membrane composition these two lipid species are exceedingly rare compared to phosphatidylethanolamine and phosphatidylglycerol headgroups. Can the authors speculate as to whether or not they think they would see similar results in their MD simulations if different lipids were initially modeled?

6. In the methods section it seems like all references to continuous wave EPR can be removed. There is no continuous wave data reported in this manuscript. Even if CW-EPR was used to check spin-label incorporation, this does not necessarily need to be included in the methods.

Minor points and grammatical errors are listed below....

- In several places throughout the manuscript the cryo-EM maps are referred to as "electron density". This is incorrect, and the maps should be referred to as coulomb potential maps or cryo-EM maps. While this may seem trivial, it is actually an important point to consider when interpreting maps to locate charged ion species near other regions of high net charge (ie: a positive charge magnesium next to three negative phosphates of an ATP)
- Introduction, line 91 – it is stated that "BmrCD is associated with antibiotic efflux". To this reviewers knowledge it has never been demonstrated that BmrCD actually transports antibiotics. While it is known that antibiotics targeting the ribosome induce BmrCD expression through a ribosome-associated transcriptional attenuation mechanism, knockout of BmrCD has been shown to have no effect on resistance to these antibiotics. Thus, whether or not BmrCD actually effluxes antibiotics, and the nature of the physiological transport substrate of BmrCD remains unknown.
- Supplemental figure 12b – it seems like D500Q should be included in this figure?
- Line 108 – remove "the" before BmrCD
- Line 202 at the end of the page – "we presume that the likely had ADP" – this sentence does not make sense.
- Line 236 – "which may then triggers the closing of" should read "which may then trigger closing of"
- Line 285, last letter of the line – "its" should be "it"

Reviewer #3:

Remarks to the Author:

This manuscript by Tang et al. details the use of various techniques to obtain the structure of BmrCD in multiple states. The manuscript is overall well written, being very thorough and clear. The main points are adequately discussed with ample citations and logical control experiments. Thus, I only have a few suggestions and questions for this manuscript.

Page 5 - You mention BmrCD is unusual because of how it differs from other type IV ABC exporters, is BmrCD type IV as well? You discuss type IV exporters on page 3, but I don't believe you ever explicitly state if BmrCD is type IV.

Page 7 - From the context of the paper, I assume that BmrCD is a heterodimer of BmrC and BmrD monomer units. Can you add a sentence that says this? Also, are there any major differences that distinguish BmrC from BmrD, or is it something small like a couple residues?

Page 9 - When testing the effect that nanodisc size has on the protein structure, you ran experiments on a 9.5nm and 12nm diameter membrane. How do these diameters compare to the approximate diameter of BmrCD within the membrane?

Page 26 - In preparing the nanodiscs + BmrCD, are you able to say if there's only one protein per nanodisc?

Page 31 - The sentence starting in line 695 sounds awkward "Followed by labeled with ...".

Page 32 - The analysis of the DEER decays is mentioned to be done with a home-written software with a reference to the Mishra et al. paper. This paper states some important details about how your analysis compares to Jeschke's DEERanalysis software. Mainly how the background subtraction in DEERanalysis can cause biases/distortions in the distance distribution output. Personally, I feel that this is important to reinclude in this manuscript, mainly because your time domain signals go out to 2-3 μ s with distances distributions centered at 6nm. Carefully discussing how you are able to reach these conclusions with relatively short DEER dipolar evolution times will solidify your claim.

Page 33 - The DEER distance distributions were generated through a sum of Gaussians while considering a statistical criterion, I'm assuming the "trust-region-reflective algorithm" mentioned in the Mishra paper. I am not entirely familiar with this algorithm, so this may be an ignorant question. Are the mean/breadth of these Gaussians held constant between each analysis with only the relative population changing, or are the DEER fit independently from one another?

REVIEWER COMMENTS

Reviewer #1 (Remarks to the Author):

In this work, the authors investigated a unique heterodimeric ABC transporter, BmrCD, using cryo-EM, DEER, and MD simulation analyses. The findings are considered to be very interesting and worthy of publication, especially the twist of ECD and the bindings of two Hoechst and Mg²⁺. However, the following revisions should be considered before publication. We thank the reviewer for the positive overall evaluation.

(1) In the context of the Introduction, the third paragraph (l.64-74) mainly refers to Pgp with two consensus sites as an example. These explanations are a little confusing. The homodimer example is a good comparison to the next paragraph on heterodimers, but even when referring to Pgp, it is necessary to clearly state that it is a transporter consisting of a single chain with two consensus sites. As a detail, ref 22 is first cited as an asymmetric example in l.68-69, but is cited as "asymmetry was not observed" in l.72-74. These are sources of confusion and require more appropriate citations and descriptions.

We have clarified the paragraph according to the reviewer's suggestions.

(2) Regarding ECD, there is no description at all before line 356 (on the second half of the results) [except for the abstract]. Although the authors may have intended to describe ABC transporters in general, the existence of ECD may be characteristic among heterodimeric transporters. The existence of ECD should be suggested at the end of the Introduction and earlier in the Results as a reason why BmrCD prefers the IF conformation (such as near l.197). The reader's understanding will change depending on whether or not they keep in mind the presence or absence of ECD.

We have noted the presence of the ECD in the introduction (see lines 121-123).

(3) One of the most difficult aspects of the authors' discussion of the transport cycle is that ATP hydrolysis appears to occur before NBD dimer formation. l.104 seems a little strange.

Considering l.455, l.104 should be modified as follows:

"demonstrating the requirement of ATP hydrolysis to stabilize the conformational intermediate, leading to the OF conformation".

We definitely did not intend to imply that.

Thank you for the suggestion. The sentence has been corrected.

(4) Related to comment 3 above, Fig. 8 of the transportation cycle may be misleading. It may be helpful to draw the top views of the NBDs on the right side (as a similar cycle) or as insets (in each). Although ATP is still bound to both sites in step 3, one is converted to ADP in step 4. It may be desirable to move the bottom figure in step 2 to step 3 (or 4).

Although it has not yet been confirmed to what extent the results match the present results, the following cycle can be assumed.

In the heterodimer, one site remains ATP-bound (s step 1). When ATP binds to the other site, it becomes unstable and begins to fluctuate (Step 2). Then, binding the two substrates further induces NBD dimerization (step 3). Dimerization is completed by binding of Mg²⁺ to the degenerate site, hydrolysis is induced at the consensus site, and the ECD is opened, leading to

the transition to OF (step 4). Finally, the two substrates are extruded, and ADP and Mg²⁺ are also dissociated, returning to step 1 and starting the next cycle.

The above would correspond to an addition of the "role of ECD" in the heterodimer transporter cycle, which can be called the transport cycle of "ECD-coupled" heterodimeric ABC transporters. In heterodimeric type transporters such as Tm287/288 and CFTR, it is believed that once ATP binds, the NBD dimer moves such that one NBS1 remains tethered (or closed) while the other NBS2 opens and closes (It may be a special case of the constant contact model proposed by Jones and George). This would be due to the fact that one ATP molecule remains bound without being hydrolyzed at the degenerate site (NBS1) (although there may be criticism). However, in BmrCD, due to the presence of ECD (its closure), not only NBS2 but also his NBS1 exhibits a very large opening behavior.

It would be helpful if this sort of thing could be shown more clearly (for general readers).

Step 2 represents a forking point where the transporter either partitions into the uncoupled cycle or bind Hoechst and continues through the transport productive cycle. We don't agree that step 2 should be moved to step 3.

Regarding the effects of the HT binding: we see no changes in the NBSs configuration hence step 3.

Regarding the NBS view, we agree. Thank you for the suggestion, we corrected it as suggested.

(5) From Figure 2, HT-2 is packed between W290 and H338 (HT-2 should be written in the text). Then, what kind of amino acid interactions stabilize HT-1, bound to both the BmrCD_IF-1HT/ATP and BmrCD_IF-1HT/ATP structures?

HT2 is added to the text as suggested. Supplementary Fig. 18 shows that aromatic residues such as F245, F293 and charged residues E246, K250, and Q384 stabilize HT1.

(6) Please add sequence alignments of heterodimeric transporters such as BmrCD, TmrAB, and TM287/288 to Supplementary Materials. Then, please explain the presence or absence of ECD and so on in the Introduction, Discussion, or other.

The sequence alignment for heterodimeric ABC transporters is added as supplementary Fig24. The ECD is also incorporated as needed.

(7) These are somewhat general questions.

(i) Is it possible to observe a state in which 1ATP and 1ADP are bound, such as when ADP and an ATP analog with low affinity are used?

(ii) To understand the general heterodimer cycle, it is possible to compare the wild type and a mutant lacking the ECD of BmrD. Are the authors considering such an approach?

Then, if it were possible, would the authors imagine it would be a cycle that favors IF as well?

(i) Yes. We used ATP and vanadate (Vi) to trap the HES state in WT BmrCD and obtained a predominantly OC structure in the particle set. It is likely had ADP and Vi at the consensus NBS, given a 10-minute incubation at 37 °C. However, we can't be sure that it is ADP in the cryo-EM maps.

(ii) We have previously constructed and reported a BmrCD variant lacking the ECD (Thaker *et al.*, Nat. Chem. Biol., 2021). Compared to WT, the mutant lacking ECD had unaltered basal ATPase activity, but the Hoechst-mediated stimulation was abrogated. Based on the structure similarity and sequence similarity between BmrCD (excluded ECD) and TmrAB (please refer to supplementary figs. 23 and 24), BmrCD without ECD may destabilize the IF in the transport

cycle.

[minor]

(8) I.35: "heterodimeric" ABC transporters & please remove "in general"

We did as suggested. Thank you.

(9) I.48-50: The first sentence of the introduction cites only a somewhat older reviews (refs 1-3). It is recommended that a recent review on humans be cited here:

Alam & Locher, *Annu Rev Biophys* 52, 275-300 (2023).

The reference is added as suggested.

(10) I.78: the catalytic glutamates "(of Walker B motif)" [as information]

Added.

(11) I.80: "heterodimeric" ABC exporters

Corrected as suggested.

(12) I.100 and through the manuscript: Hoechst or HT

I.149: BmrCD_IF-2HT/ATP

I.161: BmrCD_IF-1HT/ATP (more specific)

I.216: HT-1 & Fig.2: HT-1 and HT-2 (avoiding confusion with 1HT)

(Of course, it is also good to express only H, such as 2H, but we think it is easier to understand if it is written specifically, such as 1H.)

Very good suggestions. Corrected as suggested. We thank the reviewer.

(13) Not only the consensus site but also the degenerate site D500 of BmrC was mutated to Q. What would be the outcome if this mutation is absent? Then, please tell us the authors' intentions or the meaning of both mutations.

The rationale for replacing the D500 was to avoid the possibility of slow hydrolysis by that site as have been speculated before. We have carried out DEER analysis on the single mutant and found that the DEER data are almost identical.

(14) Although this study deals with structural transitions and substrate transport of heterodimers and performs computational studies, several recent articles and reviews are not cited. For example, we recommend citing the followings and discussing them in Discussion: Göddeke et al., *JACS* 140, 4543-4551 (2018), Göddeke & Schäfer, *JACS* 142, 12791-12801 (2020), Furuta, *Biochem Soc Trans* 49, 405-414 (2021).

We added the references.

(15) Regarding the transport cycle, in addition to ref 32 (Stockner et al), the following recent article may also be informative: Jones & George, *Int J Mol Sci* 24, 2624 (2023).

We thank the reviewer's information and added the reference.

Finally, this study appears to be a very sophisticated study of a heterodimer with an ECD. This time (due to time constraints), above, I have mainly made some fairly general comments (please forgive me if I misunderstood the authors' excellent work). Next time, I may comment on the details of the representation of data, descriptions, etc., but I hope that the revised version will be easier to understand and provide deeper insights for many readers.

We certainly welcome the reviewer's further suggestions, but we hope that the revised version addresses their concern as to further avoid publication delays.

Reviewer #2 (Remarks to the Author):

In this manuscript Tang, Q. et al. provide an extensive cryo-EM, DEER spectroscopy, and molecular dynamics based investigation on the structure and conformational changes of the ABC transporter BmrCD from *Bacillus subtilis*. Despite a wealth of recent investigations into ABC transporter structure and dynamics, key questions remain regarding the overall conformational cycle and energetic coupling mechanisms of these transporters. Questions about the asymmetric nature of ATP binding/hydrolysis persist across the field for various transporters, and especially for transporters such as BmrCD in which one of the two potential ATP binding sites is degenerate and deficient for ATP hydrolysis. Overall, the analysis of BmrCD presented here by Tang, Q. et al. sheds additional light on the nature of asymmetric ATP binding/hydrolysis in such transporters, and reveals additional details on the coupling of substrate loading in the TMDs with ATP/Mg binding and conformational transitions in the NBDs. The cryo-EM, DEER, and molecular dynamics experiments all appear to be well performed. In its current state the manuscript is quite dense with a large amount of data obtained for different BmrCD constructs under various conditions. While the density and breadth of the manuscript will make it quite difficult for the non-expert to follow, I also believe that such depth and complexity is required to support the authors claims. Moreover, the data presented in this manuscript is timely and highly relevant for all researchers interested in ABC transporter structure and function. For this reason, I only have a few comments to be addressed prior to recommending publication...

We thank the reviewer for the positive evaluation. We attempted to simplify the text without a major rewrite.

1. In this reviewer's opinion, one of the major key findings in this manuscript lies in the differential magnesium binding in the different cryo-EM structures obtained. Such a finding would mark a key aspect of BmrCD function, and would illuminate an essential component to the asymmetric nature of this transporter. However, from the figures provided (Suppl. Fig. 14) it is not apparent to me that the cryo-EM maps are of sufficient resolution and quality to support the presence or absence of magnesium. Placement of ions into cryo-EM maps of small membrane transporters is no trivial task, and I feel that the authors should better support the claims of the presence or absence of magnesium with additional figures (indicating the map threshold values) or supplemental videos that support the modeling or absence of magnesium in their final atomic models. Alternatively, the authors could provide the cryo-EM maps and atomic models to the reviewers for inspection.

Supplemental videos that support the modeling for the presence or absence of magnesium are

attached. Additional panels with different map threshold values are added in Supplementary Fig 14.

2. Along the lines of point 1 above, it is often noted in textbooks and throughout the literature that magnesium is an essential cofactor for ATP, and that “most” ATP within a cell exists pre-complexed with magnesium before Mg-ATP binds to enzymes, proteins, etc... However, in figure 8 all steps of the proposed conformational cycle show ATP binding to BmrCD before magnesium. It seems highly unlikely that such a scenario would occur in a cellular context. We agree with the reviewer that typically cellular ATP is complexed with Mg²⁺. However, our point is to highlight the differential affinities between NBSs in states as well between NBSs under *the same biochemical conditions*. These are likely to be representing transient intermediates in the transport cycle similar to the Apo conformation often reported even though cellular concentrations of ATP are 10 mM, well above the K_m. Yet we know that these states have to be populated to enable substrate binding and initiate the transport cycle.

3. In Figure 4 for spin label pairs monitoring the distance between NBS's, for the 440/441 pair it appears that the simulated distance distributions are ~10Å longer than what would be predicted based on the C-alpha positions shown in panel A. This discrepancy seems too large to be accounted for simply by spin label rotamer populations. Moreover, when incubated with H-ATP in nanodiscs (solid red trace, bottom right panel) the distance distribution for the 440/441 pair seems to accurately reflect the approximate distances indicated for a ~50/50 mixture of the IF (~45Å)/OC (~34Å) conformations observed in the cryo-EM structures. Why do the predicted distributions for the 440/441 pair seem so much longer than what is observed in the cryo-EM structures and in the experimental distribution in the presence of H-ATP?

We apologize if the calculation of the predicted distributions was not clear. We have added more details.

Briefly, distance distributions are calculated based on the cryo-EM structure using the DEER Spin-Pair Distributor at the CHARMM-GUI website (<https://www.charmm-gui.org>) with default parameters of 10 ns MD simulations with dummy spin labels that mimics the dynamics of the MTSSL spin-label. It is expected to be different from the C-alpha distance obtained by cryo-EM structures not only because of rotamers but also because the cryo-EM structure capture only one conformation in an ensemble.

4. In figure 7a and 7b it is not clear to me why in each leaflet for each conformation there are two kernel density maps? As an example, for the top left of figure 7a for the IF conformation on the upper leaflet, what is the difference between these two kernel density maps? This should be clearly indicated in the figure legend. Also, the figure/legend should be clearly amended to indicate to the reader that these kernel density maps are a view along the Z-axis perpendicular to the membrane, and the axes indicate X-Y direction in angstrom.

The figure has been replaced with a new one in the revised manuscript.

5. In the methods section on MD, it is only mentioned that BmrCD was placed into a lipid bilayer using CHARMM-GUI. What was the lipid composition of the bilayer? Was the bilayer asymmetric or symmetric? Although POPC and POPA were used in the nanodisc reconstitution, based on literature reports of *Bacillus subtilis* membrane composition these two lipid species are exceedingly rare compared to phosphatidylethanolamine and phosphatidylglycerol headgroups.

Can the authors speculate as to whether or not they think they would see similar results in their MD simulations if different lipids were initially modeled?

The composition of the lipid bilayer in the simulations was 9:1 POPC:POPA, and it was symmetric. The Tajkhorshid lab routinely simulate membrane protein systems in realistic, asymmetric lipid bilayers, sometimes composed of up to 7 types of lipids. Here, however, as the reviewer alluded to, our main objective was to reproduce the experimental conditions and the nanodisc environment used in the cryo-EM experiments. As such, we chose to model the protein in a symmetric membrane consisting of 9:1 POPC:POPA. We have revised the methods section to clarify these membrane details. The effect of largely different lipid compositions on the structure and dynamics of the protein could be rather complex, and without any data, we would prefer not to speculate too much about it, although we expect to have somewhat similar trends to be observed for neutral lipids (PE vs. PC) and for anionic lipids (PG vs. PA).

6. In the methods section it seems like all references to continuous wave EPR can be removed. There is no continuous wave data reported in this manuscript. Even if CW-EPR was used to check spin-label incorporation, this does not necessarily need to be included in the methods. We have removed the section on CW-EPR.

Minor points and grammatical errors are listed below...

- *In several places throughout the manuscript the cryo-EM maps are referred to as “electron density”. This is incorrect, and the maps should be referred to as coulomb potential maps or cryo-EM maps. While this may seem trivial, it is actually an important point to consider when interpreting maps to locate charged ion species near other regions of high net charge (ie: a positive charge magnesium next to three negative phosphates of an ATP)*

Good point. Corrected.

- *Introduction, line 91 – it is stated that “BmrCD is associated with antibiotic efflux”. To this reviewers knowledge it has never been demonstrated that BmrCD actually transports antibiotics. While it is known that antibiotics targeting the ribosome induce BmrCD expression through a ribosome-associated transcriptional attenuation mechanism, knockout of BmrCD has been shown to have no effect on resistance to these antibiotics. Thus, whether or not BmrCD actually effluxes antibiotics, and the nature of the physiological transport substrate of BmrCD remains unknown.*

The text was confusing as we meant to refer to *in vitro* data. Corrected as requested.

- *Supplemental figure 12b – it seems like D500Q should be included in this figure?*

The interactions between ATP and BmrCD_OC-ATP at the degenerate and consensus NBSs were plotted by LigPlot with the default hydrogen-bond calculation parameters setting (Maximum H-A distance is set as 2.70 Å and maximum D-A distance is set as 3.35 Å). We use this threshold to plot the figures, and for D500Q, the H-A distance and the D-A distance are 3.47 Å and 3.62 Å, which are larger than the plot threshold, so we omitted it.

- *Line 108 – remove “the” before BmrCD*

Done.

• Line 202 at the end of the page – “we presume that the likely had ADP” – this sentence does not make sense.

We corrected it as “we presume that it likely had ADP”.

• Line 236 – “which may then triggers the closing of” should read “which may then trigger closing of”

Corrected.

• Line 285, last letter of the line – “its” should be “it”

Corrected.

Reviewer #3 (Remarks to the Author):

This manuscript by Tang et al. details the use of various techniques to obtain the structure of BmrCD in multiple states. The manuscript is overall well written, being very thorough and clear. The main points are adequately discussed with ample citations and logical control experiments. Thus, I only have a few suggestions and questions for this manuscript.

Page 5 - You mention BmrCD is unusual because of how it differs from other type IV ABC exporters, is BmrCD type IV as well? You discuss type IV exporters on page 3, but I don't believe you ever explicitly state if BmrCD is type IV.

Yes. BmrCD is a member of type IV ABC exporters. At least 7 types of ABC transporters have been identified based on the TMDs folds (Thomas *et al.*, FEBS Lett., 2020), and type IV ABC exporters have a domain-swapped TMD arrangement.

Page 7 - From the context of the paper, I assume that BmrCD is a heterodimer of BmrC and BmrD monomer units. Can you add a sentence that says this? Also, are there any major differences that distinguish BmrC from BmrD, or is it something small like a couple residues?

We added a sentence in page 5 where introduce BmrCD: BmrCD is a heterodimer consisting of two protomers: BmrC and BmrD.

Page 9 - When testing the effect that nanodisc size has on the protein structure, you ran experiments on a 9.5nm and 12nm diameter membrane. How do these diameters compare to the approximate diameter of BmrCD within the membrane?

The diameter of BmrCD is about 6 nm within the membrane.

Page 26 - In preparing the nanodiscs + BmrCD, are you able to say if there's only one protein per nanodisc?

Yes. Most of the nanodiscs have one BmrCD molecule, please see the 2D images in supplementary Fig. 2. For MSP1D1 (9.5 nm), just one BmrCD can be inserted due to its 6 nm diameter. We did observe two BmrCD molecules per nanodisc in MSP1D1E3 (12 nm) in few

images. However, most of the images show one BmrCD molecule per nanodisc despite the larger diameter.

Page 31 - The sentence starting in line 695 sounds awkward "Followed by labeled with ...". We corrected this sentence.

Page 32 - The analysis of the DEER decays is mentioned to be done with a home-written software with a reference to the Mishra et al. paper. This paper states some important details about how your analysis compares to Jeschke's DEERanalysis software. Mainly how the background subtraction in DEERanalysis can cause biases/distortions in the distance distribution output. Personally, I feel that this is important to reinclude in this manuscript, mainly because your time domain signals go out to 2-3 μ s with distances distributions centered at 6nm. Carefully discussing how you are able to reach these conclusions with relatively short DEER dipolar evolution times will solidify your claim.

We have included more recent reviews and a tutorial in the references. These references illustrate the performance of the Gaussian fit approach for long distances even in short time window. The following figure is an example of the superior performance relative to DeerAnalysis. Because this included in multiple publications, we feel it would be a distraction to discuss this in the manuscript.

True P(R) (shaded region), the best-fit P(R) obtained using our Gaussian fitting approach (solid black line), and the best-fit P(R) obtained using Tikhonov regularization in DeerAnalysis (dashed purple line).

Page 33 - The DEER distance distributions were generated through a sum of Gaussians while considering a statistical criterion, I'm assuming the "trust-region-reflective algorithm" mentioned in the Mishra paper. I am not entirely familiar with this algorithm, so this may be an ignorant question. Are the mean/breadth of these Gaussians held constant between each analysis with only the relative population changing, or are the DEER fit independently from one another?

We have included an expanded description of the Global Analysis. In brief, the multiple DEER traces are fit together allowing the background and the relative populations to vary.

Reviewers' Comments:

Reviewer #1:

Remarks to the Author:

The authors have addressed almost all concerns.

However, although the authors may have already noticed, please correct the following mainly minor points.

[in MS]

l.69: "then" (or and) DEER analysis [etc.]

(There will be a conjunction before "DEER analysis" or it will end with a period.)

l.114: IF "conformation"

l.149: "substitutions" is duplicated.

(These above are likely to be fixed by the journal editing.)

Explanation of Fig. 8 (related to the previous comment 4): Let us respect the authors' opinion.

However, if we believe this cycle, it appears that ATP is hydrolyzed in step 2, independent of substrate transport.

Please describe this important point in cell biology in the manuscript, such as ``There is consumption of ATP without transport of substrates.''

[in Suppl.]

pp.37-38:

On p.37, the letters for amino acids are very blurred and difficult to read, and the tones on p.37 and p.38 are different. Moreover, both are BmrD alignments. An alignment of BmrC and its pairs (TmrB, TM287, ErfC, and TAP1) would be provided on p.37. Additionally, the ECD region is also not indicated (around residues 50-130? of BmrD).

Then, although minor, sequence alignments are generally easier to understand when comparing wild-type ones between species. When we checked the sequence, we found ...LILD"Q"... (Walker B region) in Suppl. Fig. 24, which seems to be a mutant. Please include wild type sequences such as BmrC: LILD"D" and BmrD: LILD"E", and indicate these mutational/substitutional positions with different marks, as well as Mg²⁺ coordinated residues.

Please correct these points and provide both alignments for a clearer comparison.

Furthermore, in the context of this manuscript, Mg²⁺ binding, mutations in the catalytic glutamates and the ECD are important factors. Therefore, it is appropriate for alignments to be cited from an earlier stage (by moving Suppl. Fig. 24 earlier). For example, citations at l.84 (catalytic glutamates), l.121-122 (ECD), l.132 (Mg²⁺), l.149 (D500, E592), and so on.

These modifications would make the manuscript more readable.

(In the above, comments have been kept to a minimum to avoid further publication delays.)

Finally, we thank the authors for their contributions to this field, as well as the authors and reviewers for their efforts in revising the manuscript.

Reviewer #2:

Remarks to the Author:

The authors have adequately addressed all of my questions. I recommend this manuscript for publication.

Reviewer #3:

Remarks to the Author:

I feel Mchaourab and coworkers answered the concerns outlined by the reviewers were answered in their manuscript and the work is acceptable.

REVIEWERS' COMMENTS

Reviewer #1 (Remarks to the Author):

The authors have addressed almost all concerns. However, although the authors may have already noticed, please correct the following mainly minor points.

[in MS]

I.69: "then" (or and) DEER analysis [etc.]

(There will be a conjunction before "DEER analysis" or it will end with a period.)

The sentence has been corrected.

I.114: IF "conformation"

Corrected as suggested.

I.149: "substitutions" is duplicated.

(These above are likely to be fixed by the journal editing.)

Corrected.

Explanation of Fig. 8 (related to the previous comment 4): Let us respect the authors' opinion. However, if we believe this cycle, it appears that ATP is hydrolyzed in step 2, independent of substrate transport.

Please describe this important point in cell biology in the manuscript, such as "There is consumption of ATP without transport of substrates."

Added in the Fig 8 legend.

[in Suppl.]

pp.37-38:

On p.37, the letters for amino acids are very blurred and difficult to read, and the tones on p.37 and p.38 are different. Moreover, both are BmrD alignments. An alignment of BmrC and its pairs (TmrB, TM287, ErfC, and TAP1) would be provided on p.37. Additionally, the ECD region is also not indicated (around residues 50-130? of BmrD).

Then, although minor, sequence alignments are generally easier to understand when comparing wild-type ones between species. When we checked the sequence, we found ...LILD"Q"... (Walker B region) in Suppl. Fig. 24, which seems to be a mutant. Please include wild type sequences such as BmrC: LILD"D" and BmrD: LILD"E", and indicate these mutational/substitutional positions with different marks, as well as Mg²⁺ coordinated residues. Please correct these points and provide both alignments for a clearer comparison.

We did as suggested. Thank you.

Furthermore, in the context of this manuscript, Mg²⁺ binding, mutations in the catalytic glutamates and the ECD are important factors. Therefore, it is appropriate for alignments to be

cited from an earlier stage (by moving Suppl. Fig. 24 earlier). For example, citations at I.84 (catalytic glutamates), I.121-122 (ECD), I.132 (Mg²⁺), I.149 (D500, E592), and so on. These modifications would make the manuscript more readable. (In the above, comments have been kept to a minimum to avoid further publication delays.)
Very good suggestions. Corrected as suggested. We thank the reviewer.

Finally, we thank the authors for their contributions to this field, as well as the authors and reviewers for their efforts in revising the manuscript.
We are grateful to the reviewer's efforts in helping sharpen our findings and conclusions.

Reviewer #2 (Remarks to the Author):

The authors have adequately addressed all of my questions. I recommend this manuscript for publication.
We thank the reviewer for his/her effort.

Reviewer #3 (Remarks to the Author):

I feel Mchaourab and coworkers answered the concerns outlined by the reviewers were answered in their manuscript and the work is acceptable.
We thank the reviewer for his/her effort.